# Decoupling the layers in Residual Networks

**Ricky Fok**,* **Aijun An, Zana Rashidi**
Department of Electrical Engineering and Computer Science
York University
4700 Keele Street, Toronto, M3J 1P3, Canada
`ricky.fok3@gmail.com, aan@cse.yorku.ca, zrashidi@cse.yorku.ca`

**Xiaogang Wang**
Department of Mathematics and Statistics
York University
4700 Keele Street, Toronto, M3J 1P3, Canada
`stevenw@mathstat.yorku.ca`

## Abstract

We propose a Warped Residual Network (WarpNet) using a parallelizable warp operator for forward and backward propagation to distant layers that trains faster than the original residual neural network. We apply a perturbation theory on residual networks and decouple the interactions between residual units. The resulting warp operator is a first order approximation of the output over multiple layers. The first order perturbation theory exhibits properties such as binomial path lengths and exponential gradient scaling found experimentally by Veit et al. (2016). We demonstrate through an extensive performance study that the proposed network achieves comparable predictive performance to the original residual network with the same number of parameters, while achieving a significant speed-up on the total training time. As WarpNet performs model parallelism in residual network training in which weights are distributed over different GPUs, it offers speed-up and capability to train larger networks compared to original residual networks.

## 1 Introduction

Deep Convolution Neural Networks (CNN) have been used in image recognition tasks with great success. Since AlexNet (Krizhevsky et al., 2012), many other neural architectures have been proposed to achieve start-of-the-art results at the time. Some of the notable architectures include, VGG (Simonyan & Zisserman, 2015), Inception (Szegedy et al., 2015) and Residual networks (ResNet)(He et al., 2015).

Training a deep neural network is not an easy task. As the gradient at each layer is dependent upon those in higher layers multiplicatively, the gradients in earlier layers can vanish or explode, ceasing the training process. The gradient vanishing problem is significant for neuron activation functions such as the sigmoid, where the gradient approaches zero exponentially away from the origin on both sides. The standard approach to combat vanishing gradient is to apply Batch Normalization (BN) (Ioffe & Szegedy, 2015) followed by the Rectified Linear Unit (ReLU) (Hahnloser et al., 2000) activation. More recently, skip connections (Srivastava et al., 2015) have been proposed to allow previous layers propagate relatively unchanged. Using this methodology the authors in (Srivastava et al., 2015) were able to train extremely deep networks (hundreds of layers) and about one thousand layers were trained in residual networks (He et al., 2015).

As the number of layers grows large, so does the training time. To evaluate the neural network's output, one needs to propagate the input of the network layer by layer in a procedure known as forward propagation. Likewise, during training, one needs to propagate the gradient of the loss function from the end of the network to update the model parameters, or weights, in each layer of the network using gradient descent. The complexity of forward and propagation is $O(K)$, where

---

*Corresponding Author

$K$ is the number of layers in the network. To speed up the process, one may ask if there exist a shallower network that accurately approximates a deep network so that training time is reduced. In this work we show that there indeed exists a neural network architecture that permits such an approximation, the ResNet.

Residual networks typically consist of a long chain of residual units. Recent investigations suggest that ResNets behave as an ensemble of shallow networks (Veit et al., 2016). Empirical evidence supporting this claim includes one that shows randomly deactivating residual units during training (similar to drop-out (Srivastava et al., 2014)) appears to improve performance (Huang et al., 2016). The results imply that the output of a residual unit is just a small perturbation of the input. In this work, we make an approximation of the ResNet by using a series expansion in the small perturbation. We find that merely the first term in the series expansion is sufficient to explain the binomial distribution of path lengths and exponential gradient scaling experimentally observed by Veit et al. (2016). The approximation allows us to effectively estimate the output of subsequent layers using just the input of the first layer and obtain a modified forward propagation rule. We call the corresponding operator the warp operator. The backpropagation rule is obtained by differentiating the warp operator. We implemented a network using the warp operator and found that our network trains faster on image classification tasks with predictive accuracies comparable to those of the original ResNet.

Our contributions in this work include

- We analytically investigate the properties of ResNets. In particular, we show that the first order term in the Taylor series expansion of the layer output across $K$ residual units has a binomial number of terms, which are interpreted as the number of paths in Veit et al. (2016), and that for ReLU activations the second and higher order terms in the Taylor series vanish almost exactly.

- Based on the above-mentioned analysis, we propose a novel architecture, WarpNet, which employs a warp operator as a parallelizable propagation rule across multiple layers at a time. The WarpNet is an approximation to a ResNet with the same number of weights.

- We conduct experiments with WarpNet skipping over one and two residual units and show that WarpNet achieves comparable predictive performance to the original ResNet while achieving significant speed-up. WarpNet also compares favorably with data parallelism using mini-batches with ResNet. As opposed to data parallelized ResNet where nearly all the weights are copied to all GPUs, the weights in WarpNet are distributed over various GPUs which enables training of a larger network.

The organization of this paper is as follow. In Section 2 we analyze the properties of ResNet and show that the binomial path length arises from a Taylor expansion to first order. In Section 3 we describe Warped Residual Networks. In Section 4 we show that WarpNet can attain similar performance as the original ResNet while offering a speed-up.

## 2 PROPERTIES OF RESNETS

In this section we show that recent numerical results (Veit et al., 2016) is explained when the perturbation theory is applied to ResNets. Consider the input $\mathbf{x}_i$ of the $i$-th residual unit and its output $\mathbf{x}_{i+1}$, where

$$\mathbf{x}_{i+1} = \mathbf{h}_i(\mathbf{x}_i) + \mathbf{F}_i(\mathbf{x}_i, \mathbf{W}_i). \tag{1}$$

Typically, $\mathbf{h}(\mathbf{x}_i)$ is taken to be an identity mapping, $\mathbf{h}_i(\mathbf{x}_i) = \mathbf{x}_i$. When the feature maps are down sampled, $\mathbf{h}$ is usually taken to be an $1 \times 1$ convolution layer with a stride of 2. The functions $\mathbf{F}_i$ is a combination of convolution, normalization and non-linearity layers, so that $\mathbf{W}_i$ collectively represents the weights of all layers in $\mathbf{F}_i$. In this work we only consider the case where the skip connection is the identity, $\mathbf{h}_i(\mathbf{x}_i) = \mathbf{x}_i$.

**Perturbative feature map flow** First, we show that the interpretation of ResNets as an ensemble of subnetworks is accurate up to the first order in $\mathbf{F}$ with identity mapping. One can approximate the output of a chain of residual units by a series expansion. For instance, the output of two residual units $\mathbf{x}_3$ is related to the input of the first unit by the following (we call the process where $\mathbf{x}_k$ is

expressed in terms of $\mathbf{x}_{k-1}$ an iteration. The following equations show two iterations).

$$
\begin{aligned}
\mathbf{x}_3 &= \mathbf{x}_2 + \mathbf{F}_2(\mathbf{x}_2, \mathbf{W}_2^*) \\
&= \mathbf{x}_1 + \mathbf{F}_1(\mathbf{x}_1, \mathbf{W}_1^*) + \mathbf{F}_2(\mathbf{x}_1 + \mathbf{F}_1(\mathbf{x}_1, \mathbf{W}_1^*), \mathbf{W}_2^*) \\
&= \mathbf{x}_1 + \mathbf{F}_1(\mathbf{x}_1, \mathbf{W}_1^*) + \mathbf{F}_2(\mathbf{x}_1, \mathbf{W}_2^*) + \mathbf{F}_1(\mathbf{x}_1, \mathbf{W}_1^*)\mathbf{F}_2'(\mathbf{x}_1, \mathbf{W}_2^*) + O(\epsilon^2), \quad (2)
\end{aligned}
$$

where $\mathbf{F}_2'(\mathbf{x}_1, \mathbf{W}_2^*)$ denotes the partial derivative of $\mathbf{F}_2$ with respect to $\mathbf{x}_1$ and $\mathbf{W}_i^*$ denotes the weights at the loss minimum. A Taylor series expansion in powers of $\mathbf{F}_1$ was performed on $\mathbf{F}_2$ in the second line above.[1] The $O(\epsilon^2)$ term arises from the Taylor series expansion, representing higher order terms. Equation (2) can be interpreted as an ensemble sum of subnetworks.

Below we show that the second and higher order terms are negligible, that is, the first order Taylor series expansion is almost exact, when ReLU activations are used. The second order perturbation terms all contain the Hessian $\mathbf{F}''(\mathbf{x})$. But after the network is trained, the only non-linear function in $\mathbf{F}$, ReLU, is only non-linear at the origin[2]. Therefore all second order terms vanish almost exactly. The same argument applies to higher orders.

**Theorem 1: Binomial Path Length** Let the set of indices $\sigma_c = \{c(1), c(2), \ldots c(k)\}$ obtained by choosing any subset of $S_K = \{1, 2, \cdots, K\}$, $k < K$, and then ordering such that $c(k) > c(k-1) > \cdots > c(1)$. The output across $K$ residual units with ReLU non-linearity is

$$
\mathbf{x}_{K+1} = \mathbf{x}_1 + \sum_{\sigma_c \in \mathcal{P}(S_K) \setminus \{\emptyset\}} \left( \prod_{i=2}^{k} \mathbf{F}_{c(i)}'(\mathbf{x}_1, \mathbf{W}_{c(i)}^*) \right) \mathbf{F}_{c(1)}(\mathbf{x}_1, \mathbf{W}_{c(1)}^*), \quad (3)
$$

where the sum is over all subsets $\sigma_c$ and $\mathcal{P}(S_K)$ denotes the power set of $S_K$. We have omitted the $O(\epsilon^2)$ term because the first order approximation is almost exact when ReLU is used as discussed above. The right hand side of Equation (3) is interpreted as the sum over subnetworks or paths in the sense of Veit et al. (2016). The identity path corresponding to $\sigma_c = \{\emptyset\}$ gives $\mathbf{x}_1$ in the first term. If there is only one element in $\sigma_c$, such that its cardinality $|\sigma_c| = 1$, the product on the right hand side in parentheses is absent and only terms proportional to $\mathbf{F}_{c(1)}$ appears in the sum, where $c(1) \in \{1, \ldots, K\}$. We provide the proof of Equation (3) in Appendix A.

We can make the equation simpler, solely for simplicity, by setting all weights to be the same such that $\mathbf{F}_{c(i)} = \mathbf{F}$ and $\mathbf{W}_{c(i)}^* = \mathbf{W}^*$ for all $i$,

$$
\mathbf{x}_{K+1} = \mathbf{x}_1 + \sum_{k=1}^{K} \binom{K}{k} (\mathbf{F}'(\mathbf{x}_1, \mathbf{W}^*))^{k-1} \mathbf{F}(\mathbf{x}_1, \mathbf{W}^*). \quad (4)
$$

The binomial coefficients appear because the number of subsets of $S_K$ with cardinality $k$ is $\binom{K}{k}$. Note that the implementations of our proposed method (described in Section 3) do not use this simplification.

**Exponential gradient scaling** Similarly, one observes that the gradient is the sum from all subnetwork contributions, including the identity network. The magnitudes of subnetwork gradients for an 110 layer ResNet have been measured by (Veit et al., 2016). If one takes $\mathbf{F}$ to have ReLU non-linearity, then $\mathbf{F}''(\mathbf{x}, \mathbf{W}^*) = 0$ except at the origin. The non-trivial gradient can be expressed almost exactly as $\binom{K}{k}(\mathbf{F}'(\mathbf{x}, \mathbf{W}^*))^k$. This validates the numerical results that the gradient norm decreases exponentially with subnetwork depth as reported in (Veit et al., 2016). Their experimental results indicate that the average gradient norm for each subnetwork of depth $k$ is given by $||\mathbf{F}'(\mathbf{x}, \mathbf{W}^*)||^k$.

All aforementioned properties apply only after the ResNets are trained. However, if an approximation in the network is made, it would still give similar results after training. We show in the following sections that our network can attain similar performances as the original ResNet, validating our approximation.

## 3 WARPED RESIDUAL NETWORK

The Warped Residual Network (WarpNet) is an approximation to the residual network, where $K$ consecutive residual units are compressed into one warp layer. The computation in a warp layer is

---

[1] The Taylor expansion for multivariate functions is $f(\mathbf{x} + \mathbf{a}) = f(\mathbf{x}) + \mathbf{a} \cdot \nabla_{\mathbf{x}} f(\mathbf{x}) + \cdots$

[2] Batch normalization layers are non-linear during training, due to the scaling by the sample variance.

different from that in a conventional neural network. It uses a warp operator to compute the output (i.e., $\mathbf{x}_{K+1}$) of the layer directly from the input (i.e., $\mathbf{x}_1$), as shown in Equation (4). The number of weights in a warped layer is the same of the one in the original residual network for $K$ consecutive residual units. For instance, the weights $\mathbf{W}_1, \mathbf{W}_2$ up to $\mathbf{W}_K$ are present in a warped layer. But these weights can be used and updated in parallel due to the use of the warp operator. Below we first describe the forward and backward propagation rules used in warped residual network.

## 3.1 FORWARD PROPAGATION ACROSS WARP OPERATORS

This section shows the propagation rules of the Warped Residual Network using the warp operator $T_{warp}$.

The expression for $T_{warp}$ is derived from Equation 3, that is, by using the Taylor series expansion to the first order:

$$\mathbf{x}_{K+1} = \mathbf{x}_1 + \sum_{\sigma_c \in \mathcal{P}(S_K) \backslash \{\emptyset\}} \left( \prod_{i=2}^{k} \mathbf{F}'_{c(i)}(\mathbf{x}_1, \mathbf{W}^*_{c(i)}) \right) \mathbf{F}_{c(1)}(\mathbf{x}_1, \mathbf{W}^*_{c(1)}),$$

Note that $T_{warp}$ can be calculated in a parallelizable manner for all $K$. This is shown in Figure 1 with $K = 2$, where [3]

$$\begin{aligned} \mathbf{x}_3 &= T^{K=2}_{warp}(\mathbf{x}_1, \mathbf{W}_1, \mathbf{W}_2) \\ &= \mathbf{x}_1 + \mathbf{F}_1(\mathbf{x}_1, \mathbf{W}_1) + \mathbf{F}_2(\mathbf{x}_1, \mathbf{W}_2) + \mathbf{F}'_2(\mathbf{x}_1, \mathbf{W}_2)\mathbf{F}_1(\mathbf{x}_1, \mathbf{W}_1), \end{aligned} \quad (5)$$

and $\mathbf{W}_i$ corresponds to the weights in the $i$-th residual unit in the original ResNet. The formula for the $K = 3$ case is shown in Appendix A.

## 3.2 WARPED BACK-PROPAGATION

Now we derive the backpropagation rules. Suppose that the upstream gradient $\partial L/\partial \mathbf{x}_5$ is known and we wish to compute $\partial L/\partial \mathbf{W}_1$ for gradient descent. We first back propagate the gradient down from $\mathbf{x}_5$ to $\mathbf{x}_3$. With $\mathbf{x}_5 = T_{warp}(\mathbf{x}_3)$, we can derive the backpropagated gradient

$$\begin{aligned} \frac{\partial L}{\partial \mathbf{x}_3} &= \frac{\partial L}{\partial \mathbf{x}_5} \cdot \frac{\partial \mathbf{x}_5}{\partial \mathbf{x}_3} \\ &= \frac{\partial L}{\partial \mathbf{x}_5} \cdot \left[ I + \frac{\partial \mathbf{F}_3(\mathbf{x}_3, \mathbf{W}_3)}{\partial \mathbf{x}_3} + \frac{\partial \mathbf{F}_4(\mathbf{x}_3, \mathbf{W}_4)}{\partial \mathbf{x}_3} + \frac{\partial \mathbf{F}_4(\mathbf{x}_3, \mathbf{W}_4)}{\partial \mathbf{x}_3} \frac{\partial \mathbf{F}_3(\mathbf{x}_3, \mathbf{W}_3)}{\partial \mathbf{x}_3} \right], \end{aligned}$$

where $I$ is the identity matrix and we have set the derivative of $\mathbf{F}'_4$ to zero for ReLU non-linearities. Note that we have removed all BN layers from $\mathbf{F}'_4$ in our implementation. One sees that the same kind of parallelism in the warp operator is also present for back propagation. Now we can evaluate the weight gradient for updates

$$\frac{\partial L}{\partial \mathbf{W}_1} = \frac{\partial L}{\partial \mathbf{x}_3} \cdot \frac{\partial \mathbf{x}_3}{\partial \mathbf{W}_1} = \frac{\partial L}{\partial \mathbf{x}_3} \cdot \left[ \frac{\partial \mathbf{F}_1(\mathbf{x}_1, \mathbf{W}_1)}{\partial \mathbf{W}_1} + \frac{\partial \mathbf{F}_2(\mathbf{x}_1, \mathbf{W}_2)}{\partial \mathbf{x}_1} \frac{\partial \mathbf{F}_1(\mathbf{x}_1, \mathbf{W}_1)}{\partial \mathbf{W}_1} \right].$$

Similarly for the update rule for $\mathbf{W}_2$. Rules for the all other weights in WarpNet can be obtained in the same way,

$$\frac{\partial L}{\partial \mathbf{W}_2} = \frac{\partial L}{\partial \mathbf{x}_3} \cdot \frac{\partial \mathbf{x}_3}{\partial \mathbf{W}_2} = \frac{\partial L}{\partial \mathbf{x}_3} \cdot \left[ \frac{\partial \mathbf{F}_2(\mathbf{x}_1, \mathbf{W}_2)}{\partial \mathbf{W}_2} + \frac{\partial^2 \mathbf{F}_2(\mathbf{x}_1, \mathbf{W}_2)}{\partial \mathbf{x}_1 \partial \mathbf{W}_2} \mathbf{F}_1(\mathbf{x}_1, \mathbf{W}_1) \right].$$

The weights $\mathbf{W}_1$ and $\mathbf{W}_2$ can be updated in parallel independently. The derivative $\partial \mathbf{F}_2(\mathbf{x}_1, \mathbf{W}_2)/\partial \mathbf{x}_1$ (in $\partial L/\partial \mathbf{W}_1$) is already computed in the forward pass which could be saved and reused. Furthermore, derivatives other than $\mathbf{F}'_3$ needed in $\partial L/\partial \mathbf{x}_3$ can also be computed in the forward pass. For higher warp factors $K$, only the derivative $\mathbf{F}'_{K+1}$ is not available after the forward pass.

---

[3] This equation is from Equation (2) after dropping the negligible $O(\epsilon^2)$ terms.

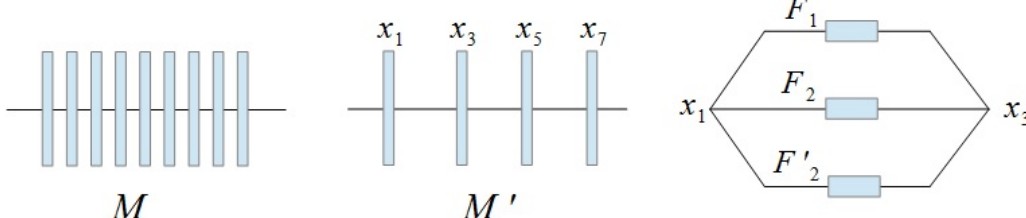

Figure 1: Forward propagation in WarpNet. Left, the original residual network where each block corresponds to a residual unit. Middle, the corresponding Warped Network with indices determined by the architecture of $M$. The diagram on the right shows the parallelism allowed in the forward pass of WarpNet. Note that $\mathbf{F}_2$ shares the same weights with $\mathbf{F}'_2$, as they come from the same residual unit in the original network $M$.

## 4  EXPERIMENTS

In this section we discuss our implementation of the WarpNet architecture and the experimental results. In order to ensure the validity of the series expansion we replace the $1 \times 1$ convolution layers on skip connections by an average pooling layer and a concatenate layer before the residual unit to reduce the spatial dimensions of feature maps and multiply their channels. In this way all skip connections are identity mappings. We adopt a wide residual architecture (WRN) (Zagoruyko & Komodakis, 2016). The convolution blocks $\mathbf{F}$ comprised of the following layers, from input to output, BN-Conv-BN-ReLU-Conv-BN (Han et al., 2016). The neural architecture of Warped Residual Networks is shown in Table 1. The layers $[T_{warp}^{(K)}] \times N_{warp}$ represent forward propagating $N_{warp}$ times, such that $\mathbf{x}_{i+K} = T_{warp}^{(K)}(\mathbf{x}_i, \mathbf{W}_i, \mathbf{W}_{i+1}, \ldots, \mathbf{W}_{i+K-1})$ and the indices $i$ correspond to the indices in the original residual network.

Using Tensorflow, we implemented a WarpNet with various parameters, $k_w$, $K$ and $N_{warp}$. The widening factor (Zagoruyko & Komodakis, 2016) is $k_w$, $K$ is the warp factor and with the scheme shown in Figure 1. We employ Tensorflow's automatic differentiation for backpropagation, where the gradients are calculated by sweeping through the network through the chain rule. Although the gradients computed in the forward pass can be re-used in the backward pass, we do not do so in our experiment and leave it to future work to potentially further speed up our method. Even so, the experimental results indicate that WarpNet can be trained faster than WRN with comparable predictive accuracy.

Consider the case $K = 2$, we found that the computation bottleneck arises from the BN layers in $\mathbf{F}'_2$. The reason being the gradient of BN layers contains an averaging operation that is expensive to compute. In our final implementation we removed all BN layers in $\mathbf{F}'_2$ from our network. This results in a departure from our series approximation but it turns out the network still trains well. This is because the normalizing layers are still being trained in $\mathbf{F}_{1,2}$. To further improve the speed-up we replace the $\mathbf{F}_1$ block in the derivative term $\mathbf{F}'_2\mathbf{F}_1$ with the input $\mathbf{x}_1$ so that the term becomes $\mathbf{F}'_2\mathbf{x}_1$. Similar approximations are made in cases where $K > 2$. We have conducted extensive experiments of this modification and found that it has similar predictive accuracies while improving speed-up. In the following, we refer to this modification of WarpNet as **WarpNet1** and the one with $\mathbf{F}'_2\mathbf{F}_1$ as **WarpNet2**. For $K = 3$ we replace all $\mathbf{F}'_j\mathbf{F}_i$ by $\mathbf{F}'_j\mathbf{x}_1$ in WarpNet1. We also drop the term $\mathbf{F}'_3\mathbf{F}'_2\mathbf{F}_1$ in computing $\mathbf{x}_4$ in both versions of WarpNet due to the limited GPUs we have in the expriements.

To investigate the speed-up provided by WarpNet and its predictive performance with various approximations on the warp operators, we define the relative speed-up, $RS$, compared to the corresponding wide residual network (WRN) as

$$RS = \frac{t_{res} - t_{warp}}{t_{res}},$$

where $t_{warp}$ is the total time to process a batch for WarpNet during training, and $t_{res}$ is that for the baseline WRN.

Table 1: Network architecture of WarpNet.

| $H \times W \times C$ | Conv blocks |
|---|---|
| $32 \times 32 \times 3$ | Input |
| $32 \times 32 \times 16$ | Conv-BN-ReLU |
| Stage 1 | Concatenate |
| $32 \times 32 \times 16k_w$ | $[T_{warp}^{(K)}] \times N_{warp}$ |
| Stage 2 | Avg_pool(2x2)-Concatenate |
| $16 \times 16 \times 32k_w$ | $[T_{warp}^{(K)}] \times N_{warp}$ |
| Stage 3 | Avg_pool(2x2)-Concatenate |
| $8 \times 8 \times 64k_w$ | $[T_{warp}^{(K)}] \times N_{warp}$ |
| $1 \times 1 \times 64k_w$ | BN-ReLU-Average_Pooling(8,8) |
|  | Fully Connected(# of classes) |

## 4.1 RESULTS ON CIFAR-10 AND CIFAR-100

For the CIFAR-10 and CIFAR-100 data sets, we trained for 80000 iterations, or 204 epochs. We took a training batch size of 128. Initial learning rate is 0.1. The learning rate drops by a factor of 0.1 at epochs 60, 120, and 160, with a weight decay of 0.0005. We use common data augmentation techniques, namely, whitening, flipping and cropping.

We study the performance of WarpNet with $K = 2$ and $K = 3$. The averaged results over two runs each are shown in Tables 2 and 3. The first column in the Tables represents the methods including two versions of WarpNet with different modifications we have made in the warp operator. A wide ResNet (WRN) is obtained by replacing $[T_{warp}^{(K)}] \times N_{warp}$ with $K \times N_{warp}$ residual units in Table 1. The total number of convolution layers (represented by $n$ in WRN-$n$-$k_w$) is $6KN_{warp} + 1$, where the factor of 6 arise from two convolution layers in each residual unit and 3 stages in the network, plus 1 convolution layer at the beginning. The number of layers in WRN is always odd as we do not use the $1 \times 1$-convolution layer across stages.

We see that in most cases, WarpNet can achieve similar, if not better, validation errors than the corresponding wide ResNet while offering speed-up. The experiments also show that the modification of replacing $\mathbf{F}'\mathbf{F}$ by $\mathbf{F}'\mathbf{x}_1$, where $\mathbf{x}_1$ is the input of the warp operator, achieves better accuracy most of the time while improving the speed-up. We observe that increasing from $K = 2$ to $K = 3$, using only one more GPU, significantly improves speed-up with only a slight drop in validation accuracy compared to the $K = 2$ case.

We have also performed experiments on the speed-up as the widening factor $k_w$ increases. We found that the speed-up increases as the WarpNet gets wider. For $k_w = 4, 8$ and 16, the speed-up in total time for $K = 2$ is $35\%, 40\%$ and $42\%$ respectively. The speed-up also increases with the warp factor $K$, for $K = 3$ using the $\mathbf{F}'\mathbf{x}$ modification, the speed-ups are $44\%, 48\%$ and $50\%$ respectively.

## 4.2 RESULTS ON IMAGENET

We also tested WarpNet on a down-sampled (32x32) ImageNet data set (Chrabaszcz & Hutter, 2017). The data set contains 1000 classes with 1281167 training images and 50000 validation images with 50 images each class. The training batch size is 512, initial learning rate is 0.4 and drops by a factor of 0.1 at every 30 epochs. The weight decay is set to be 0.0001. We use the overall best performing warp operator in the CIFAR experiments, namely, the one containing $\mathbf{F}'\mathbf{x}$.

The results are shown in Table 4 and Figure 2. First, we show directly that for a given ResNet there exists a WarpNet that obtains a higher validation accuracy with shorter training time. We increase $K$ from 2 to 3 and keep everything else fixed. This corresponds to WarpNet-109-2. The network has more residual units than WRN-73-2. We observed that WarpNet-109-2 trains 12% faster than WRN-73-2 while resulting in a better validation accuracy. Second, WarpNet can achieve close to the benchmark validation error of 18.9% with WRN-28-10 in (Chrabaszcz & Hutter, 2017). Note that we were not able to train the corresponding WRN-73-4 on the dataset as the model requires too much memory on a single GPU. This shows that the weight distribution of WarpNet across GPUs

Table 2: Validation error and relative speed-up in parentheses $(\cdots)$ of WarpNet with $K = 2$. Brackets $[\cdots]$ under "GPU assignment" indicates a GPU is used to compute (and store the weights needed by) the operation enclosed. 3 GPUs are used in this experiment. The corresponding WRN with the same number of parameters is listed in the last row in each block.

| CIFAR-10, $N_{warp} = 3$ | GPU assignment | $k_w = 4$ | $k_w = 6$ |
|---|---|---|---|
| WarpNet1-37-$k_w$ | $[\mathbf{F}_1], [\mathbf{F}_2], [\mathbf{F}'_2\mathbf{x}_1]$ | 5.18 (33%) | **4.79** (36%) |
| WarpNet2-37-$k_w$ | $[\mathbf{F}_1], [\mathbf{F}_2], [\mathbf{F}'_2\mathbf{F}_1]$ | 5.15 (23%) | 4.92 (26%) |
| WRN-37-$k_w$ | $[\mathbf{F}_1, \mathbf{F}_2]$ | **4.97** (0%) | 5.01 (0%) |
| CIFAR-10, $N_{warp} = 6$ | GPU assignment | $k_w = 4$ | $k_w = 6$ |
| WarpNet1-73-$k_w$ | $[\mathbf{F}_1], [\mathbf{F}_2], [\mathbf{F}'_2\mathbf{x}_1]$ | **4.51** (34%) | 4.76 (36%) |
| WarpNet2-73-$k_w$ | $[\mathbf{F}_1], [\mathbf{F}_2], [\mathbf{F}'_2\mathbf{F}_1]$ | 4.86 (24%) | 4.91 (26%) |
| WRN-73-$k_w$ | $[\mathbf{F}_1, \mathbf{F}_2]$ | 4.80 (0%) | **4.71** (0%) |
| CIFAR-100, $N_{warp} = 3$ | GPU assignment | $k_w = 4$ | $k_w = 6$ |
| WarpNet1-37-$k_w$ | $[\mathbf{F}_1], [\mathbf{F}_2], [\mathbf{F}'_2\mathbf{x}_1]$ | **22.4** (33%) | **21.6** (36%) |
| WarpNet2-37-$k_w$ | $[\mathbf{F}_1], [\mathbf{F}_2], [\mathbf{F}'_2\mathbf{F}_1]$ | 22.8 (23%) | 22.0 (26%) |
| WRN-37-$k_w$ | $[\mathbf{F}_1, \mathbf{F}_2]$ | 22.8 (0%) | **21.6** (0%) |
| CIFAR-100, $N_{warp} = 6$ | GPU assignment | $k_w = 4$ | $k_w = 6$ |
| WarpNet1-73-$k_w$ | $[\mathbf{F}_1], [\mathbf{F}_2], [\mathbf{F}'_2\mathbf{x}_1]$ | **21.4** (34%) | **21.1** (36%) |
| WarpNet2-73-$k_w$ | $[\mathbf{F}_1], [\mathbf{F}_2], [\mathbf{F}'_2\mathbf{F}_1]$ | 21.7 (24%) | 21.4 (26%) |
| WRN-73-$k_w$ | $[\mathbf{F}_1, \mathbf{F}_2]$ | 21.8 (0%) | 21.9 (0%) |

Table 3: Validation error and relative speed-up in parentheses $(\cdots)$ of WarpNet with $K = 3$. Brackets $[\cdots]$ under "GPU assignment" indicates a GPU is used to compute (and store the weights needed by) the operation enclosed. 4 GPUs are used in this experiment. The corresponding WRN with the same number of parameters is listed in the last row.

| CIFAR-10, $N_{warp} = 2$ | GPU assignment | $k_w = 4$ | $k_w = 6$ |
|---|---|---|---|
| WarpNet1-37-$k_w$ | $[\mathbf{F}_1], [\mathbf{F}_2], [\mathbf{F}_3], [\mathbf{F}'_2\mathbf{x}_1, \mathbf{F}'_3\mathbf{x}_1]$ | 5.39 (43%) | 5.35 (47%) |
| WarpNet2-37-$k_w$ | $[\mathbf{F}_1], [\mathbf{F}_2], [\mathbf{F}_3], [\mathbf{F}'_2\mathbf{F}_1, \mathbf{F}'_3\mathbf{F}_1, \mathbf{F}'_3\mathbf{F}_2]$ | 5.52 (34%) | 5.22 (33%) |
| WRN-37-$k_w$ | $[\mathbf{F}_1, \mathbf{F}_2, \mathbf{F}_3]$ | **4.97** (0%) | **5.01** (0%) |
| CIFAR-10, $N_{warp} = 4$ | GPU assignment | $k_w = 4$ | $k_w = 6$ |
| WarpNet1-73-$k_w$ | $[\mathbf{F}_1], [\mathbf{F}_2], [\mathbf{F}_3], [\mathbf{F}'_2\mathbf{x}_1, \mathbf{F}'_3\mathbf{x}_1]$ | **4.66** (45%) | **4.64** (46%) |
| WarpNet2-73-$k_w$ | $[\mathbf{F}_1], [\mathbf{F}_2], [\mathbf{F}_3], [\mathbf{F}'_2\mathbf{F}_1, \mathbf{F}'_3\mathbf{F}_1, \mathbf{F}'_3\mathbf{F}_2]$ | 4.81 (34%) | 4.77 (34%) |
| WRN-73-$k_w$ | $[\mathbf{F}_1, \mathbf{F}_2, \mathbf{F}_3]$ | 4.80 (0%) | 4.71 (0%) |
| CIFAR-100, $N_{warp} = 2$ | GPU assignment | $k_w = 4$ | $k_w = 6$ |
| WarpNet1-37-$k_w$ | $[\mathbf{F}_1], [\mathbf{F}_2], [\mathbf{F}_3], [\mathbf{F}'_2\mathbf{x}_1, \mathbf{F}'_3\mathbf{x}_1]$ | **22.8** (43%) | 22.2 (47%) |
| WarpNet2-37-$k_w$ | $[\mathbf{F}_1], [\mathbf{F}_2], [\mathbf{F}_3], [\mathbf{F}'_2\mathbf{F}_1, \mathbf{F}'_3\mathbf{F}_1, \mathbf{F}'_3\mathbf{F}_2]$ | 23.0 (34%) | 23.1 (33%) |
| WRN-37-$k_w$ | $[\mathbf{F}_1, \mathbf{F}_2, \mathbf{F}_3]$ | **22.8** (0%) | **21.6** (0%) |
| CIFAR-100, $N_{warp} = 4$ | GPU assignment | $k_w = 4$ | $k_w = 6$ |
| WarpNet1-73-$k_w$ | $[\mathbf{F}_1], [\mathbf{F}_2], [\mathbf{F}_3], [\mathbf{F}'_2\mathbf{x}_1, \mathbf{F}'_3\mathbf{x}_1]$ | 22.0 (45%) | 21.4 (46%) |
| WarpNet2-73-$k_w$ | $[\mathbf{F}_1], [\mathbf{F}_2], [\mathbf{F}_3], [\mathbf{F}'_2\mathbf{F}_1, \mathbf{F}'_3\mathbf{F}_1, \mathbf{F}'_3\mathbf{F}_2]$ | 21.9 (34%) | **21.2** (35%) |
| WRN-73-$k_w$ | $[\mathbf{F}_1, \mathbf{F}_2, \mathbf{F}_3]$ | **21.8** (0%) | 21.9 (0%) |

allows a bigger network to be trained. Remarkably, the validation error curve for WRN-73-2 and its approximation WarpNet 73-2 ($K = 2, N_{warp} = 6$) lie almost exactly on top of each other. This suggests that our implementation of WarpNet is a good approximation of the corresponding WRN *throughout* training.

## 4.3 COMPARISON WITH DATA PARALLELISM

WarpNet offers model parallelism to ResNet learning, in which different sets of weights are learned in parallel. In comparison, a popular way to parallelize deep learning is to split the batch in each training iteration into subsets and allow a different GPU to compute gradients for all weights based on a different subset and synchronization can be done, e.g., by averaging the gradients from all GPUs and updating the weights based on the average. We refer to such methods as data parallelism methods. Below we compare WarpNet with a data parallelism method on 2 or 4 GPUs on CIFAR-10

Table 4: Top-5 validation errors (%) on ImageNet32×32 and the relative speed-up compared to WRN-73-2 in parentheses for $k_w = 2$. The warp operator containing $\mathbf{F'x}$ is used for WarpNet.

|  | # of parameters | ImageNet32 × 32 |
| --- | --- | --- |
| WRN-73-2 | 4.8M | 23.0 (0%) |
| WarpNet1-73-2 ($K = 2, N_{warp} = 6$) | 4.8M | 23.0 (27%) |
| WarpNet1-73-2 ($K = 3, N_{warp} = 4$) | 4.8M | 23.5 (41%) |
| WarpNet1-109-2($K = 3, N_{warp} = 6$) | 7.1M | **22.1** (12%) |
| WarpNet1-73-4 ($K = 2, N_{warp} = 6$) | 18.9M | 19.5 |
| WRN-28-10 (Chrabaszcz & Hutter, 2017) | 37.1M | **18.9** |

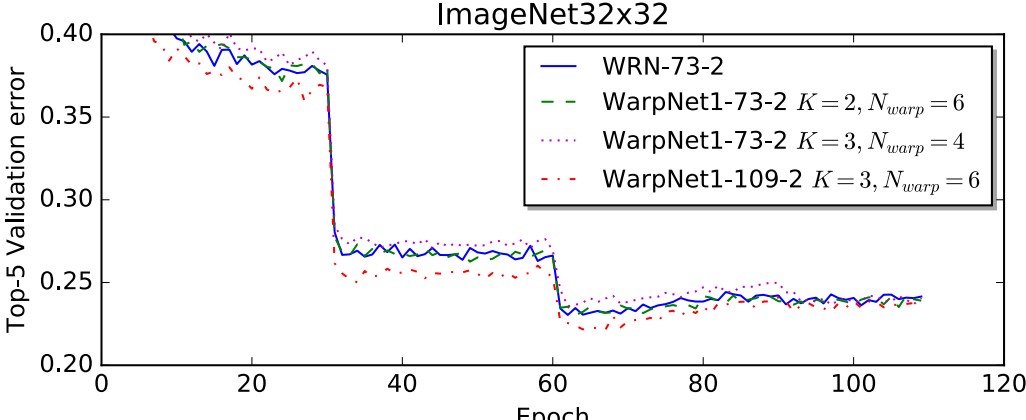

Figure 2: Top-5 validation error of the down-sampled ImageNet data set for $k_w = 2$.

for which we divide each batch into 2 or 4 mini-batches, respectively, and synchronization is done right after all GPUs finish their job on their mini-batch to avoid harming the accuracy. Table 5 shows the average result over 2 runs for each method. All methods see the same volume of data during training, which means that the number of epochs is the same for all methods. We chose the warp operators containing $\mathbf{F'x}$ in this experiment, that is, WarpNet1 whose operations are specified in the first rows of each block in Tables 2 and 3. We use the GPU assignment $[\mathbf{F}_1], [\mathbf{F}_2, \mathbf{F'}_2\mathbf{x}_1], [\mathbf{F}_3, \mathbf{F'}_3\mathbf{x}_1]$ for the case with 3 GPUs.

The results show that WarpNet is more accurate than data parallelism in both 2-GPU and 4-GPU cases. When 3 or 4 GPUs are used, WarpNet is much faster than data-parallelized ResNet with 4 GPUs. We believe this is because the data parallelism method needs to store all the weights of the model in all GPUs and its speed is slowed by the need to update all the weights across all GPUs at the time of synchronization. In comparison, WarpNet splits the weights among GPUs and each GPU only maintains and updates a subset of weights. Such weight distributions in WarpNet require less GPU memory, which allows it to train larger networks. Furthermore, data parallelism can be applied to WarpNet as well to potentially further speed up WarpNet, which is a topic beyond the scope of this paper.

## 5 CONCLUDING REMARKS

In this paper, we proposed the Warped Residual Network (WarpNet) that arises from the first order Taylor series expansion with ReLU non-linearity. We showed analytically that the first order expansion is sufficient to explain the ensemble behaviors of residual networks (Veit et al., 2016). The Taylor series approximation has the structure that allows WarpNet to train consecutive residual units in parallel while ensuring that the performance is similar to the corresponding ResNet. The weights of different residual units are distributed over the vairous GPUs which enables the training of bigger networks compared to ResNets given limited GPU memory. Experimental results show that Warp-Net can provide a significant speed-up over wide ResNets with similar predictive accuracy, if not

Table 5: Comparison with data parallelism.

| | # of GPUs | Validation error (%) | Relative Speed-up |
|---|---|---|---|
| WRN-73-4 | 1 | 4.80 | 0% |
| WRN-73-4 | 2 | 4.86 | 29.6% |
| WRN-73-4 | 4 | 4.67 | 32.4% |
| WarpNet1-73-4, $K = 2, N_{warp} = 6$ | 2 | **4.51** | 23.0% |
| WarpNet1-73-4, $K = 3, N_{warp} = 4$ | 3 | 4.66 | 44.1% |
| WarpNet1-73-4, $K = 3, N_{warp} = 4$ | 4 | 4.66 | **44.6**% |

better. We also show that WarpNet outperforms a data parallelism method on ResNet, achieving better predictive accuracies and a much better speed up when more than 2 GPUs are used.

ACKNOWLEDGMENTS

We thank Wenxin Xu for providing his code for ResNet at `https://github.com/wenxinxu/resnet_in_tensorflow`.

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

## A  Binomial path length

In this section we explicitly work out the expressions for $\mathbf{x}_3$ and $\mathbf{x}_4$ using the Taylor expansion and show that in the general case the path lengths $k$ corresponds to the binomial number of terms with power $k$ in $\mathbf{F}$ and $\mathbf{F}'$ together in the first order Taylor expansion. The terms of order $O(\epsilon^2)$ will be omitted in this section. The expression for $\mathbf{x}_3$ is

$$
\begin{aligned}
\mathbf{x}_3 &= \mathbf{x}_2 + \mathbf{F}_2(\mathbf{x}_2, \mathbf{W}_2^*) \\
&= \mathbf{x}_1 + \mathbf{F}_1(\mathbf{x}_1, \mathbf{W}_1^*) + \mathbf{F}_2(\mathbf{x}_1 + \mathbf{F}_1(\mathbf{x}_1, \mathbf{W}_1^*), \mathbf{W}_2^*).
\end{aligned}
\tag{6}
$$

Taylor expanding the last term in powers of $\mathbf{F}_1$ gives

$$
\begin{aligned}
\mathbf{x}_3 &= \mathbf{x}_1 + \mathbf{F}_1(\mathbf{x}_1, \mathbf{W}_1^*) + \mathbf{F}_2(\mathbf{x}_1, \mathbf{W}_2^*) + \left.\frac{\partial \mathbf{F}_2(\mathbf{x}, \mathbf{W}_2^*)}{\partial \mathbf{x}}\right|_{\mathbf{x}=\mathbf{x}_1} \mathbf{F}_1(\mathbf{x}_1, \mathbf{W}_1^*) \\
&= \mathbf{x}_1 + \mathbf{F}_1(\mathbf{x}_1, \mathbf{W}_1^*) + \mathbf{F}_2(\mathbf{x}_1, \mathbf{W}_2^*) + \mathbf{F}_2'(\mathbf{x}_1, \mathbf{W}_2^*)\mathbf{F}_1(\mathbf{x}_1, \mathbf{W}_1^*),
\end{aligned}
\tag{7}
$$

where in the last equality we simplified the notation for the partial derivative, where $\partial/\partial\mathbf{x} = (\partial/\partial x_1, \ldots, \partial/\partial x_D)$ and $D$ is the dimensionality of $\mathbf{x}$. Counting the powers of $\mathbf{F}$ and $\mathbf{F}'$ reveals that there are (1,2,1) terms for each power 0, 1 and 2, respectively. The same (1,2,1) coefficients can also be obtained by setting the weights to be the same $\mathbf{W}_i^* = \mathbf{W}_j^*$ for all $i, j$.

For $\mathbf{x}_4$,

$$
\begin{aligned}
\mathbf{x}_4 &= \mathbf{x}_3 + \mathbf{F}_3(\mathbf{x}_3, \mathbf{W}_3^*) \\
&= \mathbf{x}_2 + \mathbf{F}_2(\mathbf{x}_2, \mathbf{W}_2^*) + \mathbf{F}_3(\mathbf{x}_2 + \mathbf{F}_2(\mathbf{x}_2, \mathbf{W}_2^*), \mathbf{W}_3^*) \\
&= \mathbf{x}_2 + \mathbf{F}_2(\mathbf{x}_2, \mathbf{W}_2^*) + \mathbf{F}_3(\mathbf{x}_2, \mathbf{W}_3^*) + \mathbf{F}_3'(\mathbf{x}_2, \mathbf{W}_3^*)\mathbf{F}_2(\mathbf{x}_2, \mathbf{W}_2^*).
\end{aligned}
\tag{8}
$$

This is similar to $\mathbf{x}_3$ but with indices on the right hand side increased by 1. One more iteration of Taylor expansion gives $\mathbf{x}_4$ in terms of $\mathbf{x}_1$

$$
\begin{aligned}
\mathbf{x}_4 = \ &\mathbf{x}_1 \\
+\ &\mathbf{F}_1(\mathbf{x}_1, \mathbf{W}_1^*) + \mathbf{F}_2(\mathbf{x}_1, \mathbf{W}_2^*) + \mathbf{F}_3(\mathbf{x}_1, \mathbf{W}_3^*) \\
+\ &\mathbf{F}_2'(\mathbf{x}_1, \mathbf{W}_3^*)\mathbf{F}_1(\mathbf{x}_1, \mathbf{W}_1^*) + \mathbf{F}_3'(\mathbf{x}_1, \mathbf{W}_3^*)\mathbf{F}_1(\mathbf{x}_1, \mathbf{W}_1^*) + \mathbf{F}_3'(\mathbf{x}_1, \mathbf{W}_3^*)\mathbf{F}_2(\mathbf{x}_1, \mathbf{W}_2^*) \\
+\ &\mathbf{F}_3'(\mathbf{x}_1, \mathbf{W}_3^*)\mathbf{F}_2'(\mathbf{x}_1, \mathbf{W}_2^*)\mathbf{F}_1(\mathbf{x}_1, \mathbf{W}_1^*).
\end{aligned}
\tag{9}
$$

where we have organized all terms having the same power of $\mathbf{F}$ and $\mathbf{F}'$ together to be in the same row. We also assume ReLU is used so that $\mathbf{F}_3'' = 0$ almost exactly. We say that a term in the first order expansion has power $k$ if the term is proportional to $(\mathbf{F}')^{k-1}\mathbf{F}$. Then there are (1,3,3,1) terms for each power $k \in \{1, 2, 3, 4\}$. A pattern begins to emerge that the number of terms for each power of $\mathbf{F}$ satisfy $\binom{K}{k}$, where $K$ is the number skipped, i.e. $K = 3$ for the $\mathbf{x}_4$ to $\mathbf{x}_1$ case above.

Now we show that the number of terms in the first order expansion is the binomial coefficient for all $k$. We aim to derive a recursion relationship between each iteration of *index reduction*. We define the index reduction as operations that reduce the index of the outputs $\mathbf{x}_i$ by one. For instance, residual unit formula $\mathbf{x}_i = \mathbf{x}_{i-1} + \mathbf{F}_{i-1}$ is an index reduction, where the index is reduced from $i$ to $i-1$. Note that this operation generates a term of power 1, $\mathbf{F}_{i-1}$, from a power 0 term $\mathbf{x}_i$. The first order Taylor expansion generates a term of an additional power with a derivative,

$$
\mathbf{F}_i(\mathbf{x}_i) = \mathbf{F}_i(\mathbf{x}_{i-1} + \mathbf{F}_{i-1}(\mathbf{x}_{i-1})) = \mathbf{F}_i(\mathbf{x}_{i-1}) + \mathbf{F}_i'(\mathbf{x}_{i-1})\mathbf{F}_{i-1}(\mathbf{x}_{i-1}),
$$

where an index reduction is used in the first equality and the Taylor expansion is used in the second. The dependence on $\mathbf{F}$ upon the weights and higher order corrections are omitted to avoid clutter. We see the the combination of an index reduction and Taylor expansion generate terms of powers $k$ and $k + 1$ with index $i - 1$ from a term of power $k$ of index $i$. Let $C(K, k)$ be the number of terms of $K$ index reduction operations and power $k$. For instance, $K = 3$ corresponds to expressing $\mathbf{x}_4$ in terms of $\mathbf{x}_1$ as in Equation 9 with $C(3, 1) = C(3, 2) = 3$ and $C(3, 0) = C(3, 3) = 1$.

### A.1  First Proof of Theorem 1, Recursion Relations

We now derive a relationship between the number of terms of power $k + 1$ after $K + 1$ index reductions with those after $K$ index reductions. Consider the terms corresponding to $K + 1$ with

power $k + 1$. There are two sources of such terms. First, those generated by an additional index reduction after $K$ operations and the zeroth order Taylor expansion in terms of power $k + 1$, there are $C(K, k + 1)$ such terms. Second, those generated by the first order Taylor expansion in terms of power $k$, there are $C(K, k)$ such terms. Therefore the total number of terms with power $k + 1$ after $K + 1$ index reductions is

$$C(K + 1, k + 1) = C(K, k + 1) + C(K, k).$$

This is precisely the recursion formula satisfied by the binomial coefficients. We have explicitly shown earlier that for $K = 3$ and $K = 4$ the coefficients are binomial coefficients. Therefore the number of terms at any $K$ and power $k$ are the binomial coefficients, $C(K, k) = \binom{K}{k}$.

Note that the order of the indices in $[\Pi_{i=2}^{K} \mathbf{F}'_{c(i)}]\mathbf{F}_{c(1)}$ must be ordered, such that $c(k) > c(k - 1) > \cdots > c(1)$. Where the indices $c(i, k)$ are given by any subset of the integers $S_K = \{1, 2, \ldots, K\}$. Of course, the number of unordered subsets with cardinality $k$ from a set of cardinality $K$ is $\binom{K}{k}$. To write down a term of power $k$ explicitly in the first order Taylor expansion, we first choose a unordered subset of $k$ indices from $S_K$ then we order the indices to form $\sigma_c = \{c(k), \ldots, c(1)\}$. Then the output after $K$ residual units with input $\mathbf{x}_i$ is the sum over all these subsets

$$\mathbf{x}_{K+1} = \sum_{\sigma_c \in \mathcal{P}(S_K)} \left( \prod_{i=2}^{k} \mathbf{F}'_{c(i)}(\mathbf{x}_1, \mathbf{W}^*_{c(i)}) \right) \mathbf{F}_{c(1)}(\mathbf{x}_1, \mathbf{W}^*_{c(1)}), \tag{10}$$

where $\mathcal{P}(S_K)$ denotes the power set of $S_K$. Note that when $\sigma_c$ is empty, the right hand side gives the identity mapping. This is the same as Equation (3). Setting all weights to be the same gives the form in Equation 4. $\square$

## A.2   SECOND PROOF OF THEOREM 1, BERNOULLI PROCESS

The series of index reduction operations can be identified with a Bernoulli process with parameters $K$ and $p = 0.5$. Each term in Equation (3) arises from a realization of the Bernoulli process. Summing over terms from all possible realizations results in Equation (3). Recall that to express $\mathbf{x}_{K+1}$ in terms of $\mathbf{x}_1$ similar to Equation (3), we need $K$ index reduction operations. Let $X_{K:1} := \{X_K, X_{K-1}, \ldots, X_1\}$ be a Bernoulli process, where $X_i \sim \mathcal{B}(K, p = 0.5)$. Then the realizations $X_i = 0$ represents the power of a term remains the same after an index reduction, and $X_i = 1$ denotes an increase in the power of a term by one. For example, consider $K = 2$, the terms corresponding to the realizations of the Bernoulli process $X_{2:1} = \{X_2, X_1\}$ are

$$
\begin{aligned}
\{0, 0\} &\rightarrow \mathbf{x}_1 \\
\{1, 0\} &\rightarrow \mathbf{F}_2 \\
\{0, 1\} &\rightarrow \mathbf{F}_1 \\
\{1, 1\} &\rightarrow \mathbf{F}'_2 \mathbf{F}_1
\end{aligned}
$$

One sees that $\mathbf{x}_3$ can be obtained by summing over all terms corresponding to all realizations of $X_{3:1}$. This generalizes to $X_{K:1}$ for $\mathbf{x}_{K+1}$. The probability of a term having power $k$ is $2^{-K} \binom{K}{k}$. Since the total number of terms is $2^K$, the number of terms having power $k$ is the binomial coefficient $\binom{K}{k}$. If we let $\sigma_c$ to be the term corresponding to a realization of $X_{K:1}$, then consecutive Taylor expansions corresponds to summing over all $\sigma_c$ and Equation (3) follows. $\square$

