# OpenReview forum: "Decoupling the Layers in Residual Networks"
_ICLR.cc/2018/Conference — Accept (Poster)_

### Official Review · AnonReviewer1 · 2017-11-27
**warp operator**

**Rating:** 6
**Confidence:** 3

**Review:**

Motivated via Talor approximation of the Residual network on a local minima, this paper proposed a warp operator that can replace a block of a consecutive number of residual layers. While having the same number of parameters as the original residual network, the new operator has the property that the computation can be parallelized. As demonstrated in the paper, this improves the training time with multi-GPU parallelization, while maintaining similar performance on CIFAR-10 and CIFAR-100.

One thing that is currently not very clear to me is about the rotational symmetry. The paper mentioned rotated filters, but continue to talk about the rotation in the sense of an orthogonal matrix applying to the weight matrix of a convolution layer. The rotation of the filters (as 2D images or images with depth) seem to be quite different from "rotating" a general N-dim vectors in an abstract Euclidean space. It would be helpful to make the description here more explicit and clear.

---

> ### Author Response · Authors · 2018-01-03
> **Response to AnonReviewer1**
>
> Dear AnonReviewer1,
>
>  Thank you for your comments. We are sorry about the confusion in our discussion. We decided to remove the discussion on symmetry breaking and use the space to present a much more extensive experimental study of WarpNet on CIFAR-10 and CIFAR-100, an analysis on ImageNet, a scale-up of the warp factor from K=2 to K=3, and a comparison to data parallelism. In addition, we clarified the notations in derivatives and added a theorem and its proof (in Appendix). We have posted the new version of the paper. We do feel that the paper is much stronger than before, thanks to the reviewers’ suggestions."

---

### Official Review · AnonReviewer3 · 2017-11-27
**Creative investigation of Resnets**

**Rating:** 7
**Confidence:** 3

**Review:**

Paper proposes a shallow model for approximating stacks of Resnet layers, based on mathematical approximations to the Resnet equations and experimental insights, and uses this technique to train Resnet-like models in half the time on CIFAR-10 and CIFAR-100. While the experiments are not particularly impressive, I liked the originality of this paper.

---

> ### Author Response · Authors · 2018-01-03
> **Response to AnonReviewer3**
>
> Dear AnonReviewer3,
>
> Thank you for your support! In this revision we have included extensive experimental results on CIFAR-10, CIFAR-100, and ImageNet data sets using various settings for WarpNet, including increasing the warp factor from K=2 to K=3. We also compared our parallelization with data parallelization using mini-batches for ResNets. Please see the new experimental results in Section 4.

---

### Official Review · AnonReviewer2 · 2017-11-29
**Review of "Decoupling the Layers in Residual Networks"**

**Rating:** 6
**Confidence:** 3

**Review:**

The main contribution of this paper is a particular Taylor expansion of the outputs of a ResNet which is shown to be exact at almost all points in the input space.  This expression is used to develop a new layer called a “warp layer” which essentially tries to compute several layers of the residual network using the Taylor expansion expression — however in this expression, things can be done in parallel, and interestingly, the authors show that the gradients also decouple when the (ResNet) model is close to a local minimum in a certain sense, which may motivate the decoupling of layers to begin with.  Finally the authors stack these warp layers to create a “warped resnet” which they show does about as well as an ordinary ResNet but has better parallelization properties.

To me the analytical parts of the paper are the most interesting, particularly in showing how the gradients approximately decouple.  However there are several weaknesses to the paper (or maybe just things I didn’t understand).  First,  a major part of the paper tries to make the case that there is a symmetry breaking property of the proposed model, which I am afraid I simply was not able to follow.  Some of the notation is confusing here — for example, presumably the rotations refer to image level rotations rather than literally multiplying the inputs by an orthogonal matrix, which the notation suggests to be the case.  It is also never precisely spelled out what the final theoretical guarantee is (preferably the authors would do this in the form of a proposition or theorem).

Throughout, the authors write out equations as if the weights in all layers are equal, but this is confusing even if the authors say that this is what they are doing, since their explanation is not very clear.  The confusion is particularly acute in places where derivatives are taken, because the derivatives continue to be taken as if the weights were untied, but then written as if they happened to be the same.

Finally the experimental results are okay but perhaps a bit preliminary.  I have a few recommendations here:
* It would be stronger to evaluate results on a larger dataset like ILSVRC.
* The relative speed-up of WarpNet compared to ResNet needs to be better explained — the authors break the computation of the WarpNet onto two GPUs, but it’s not clear if they do this for the (vanilla) ResNet as well.  In batch mode, the easiest way to parallelize is to have each GPU evaluate half the batch.  Even in a streaming mode where images need to be evaluated one by one, there are ways to pipeline execution of the residual blocks, and I do not see any discussion of these alternatives in the paper.
* In the experimental results, K is set to be 2, and the authors only mention in passing that they have tried larger K in the conclusion.  It would be good to have a more thorough experimental evaluation of the trade-offs of setting K to be higher values.

A few remaining questions for the authors:
* There is a parallel submission (presumably by different authors called “Residual Connections Encourage Iterative Inference”) which contains some related insights.  I wonder what are the differences between the two Taylor expansions, and whether the insights of this paper could be used to help the other paper and vice versa?
* On implementation - the authors mention using Tensorflow’s auto-differentiation.  My question here is — are gradients being re-used intelligently as suggested in Section 3.1?
* I notice that the analysis about the vanishing Hessian could be applied to most of the popular neural network architectures available now.  How much of the ideas offered in this paper would then generalize to non-resnet settings?

---

> ### Author Response · Authors · 2018-01-03
> **Response to AnonReviewer2**
>
> Dear AnonReviewer2,
>
> Thank you for your constructive comments. We have revised the paper accordingly, considering all your suggestions. In particular, we conducted a much more extensive experimental study of WarpNet, clarified the notations in derivatives and added a theorem and its proof. Below we describe how we addressed each of your points:
>
> *Throughout, the authors write out equations as if the weights in all layers are equal, but this is confusing even if the authors say that this is (not) what they are doing, since their explanation is not very clear.  The confusion is particularly acute in places where derivatives are taken, because the derivatives continue to be taken as if the weights were untied, but then written as if they happened to be the same.
>
> We have added a general formula for the first order Taylor series for all K in Equation (3) to clarify how the equation should be read. The exponential gradient scaling result can then be derived by differentiating this expression with respect to x. The only non vanishing term in the differentiation comes from the right-most factor, F, since all F'' = 0 almost exactly when ReLU is used. Then setting all weights to be equal results in the binomial coefficient and (F') to the power of k. We have also clarified the derivation of the formula in the gradient decoupling paragraph. In addition, we have provided a proof of binomial path lengths in the appendix and we hope that this will clarify the presentation of this paper.
>
> * It would be stronger to evaluate results on a larger dataset like ILSVRC.
>
> We have added a subsection (4.2) that compares WarpNet with ResNet on ImageNet on a few different settings. Using the ImageNet data set, we also illustrates the "almost-exactness" of the first order Taylor series in Figure 2, where the validation curves of the WRN-73-2 and it's approximation WarpNet-73-2 (K=2) approximation lie almost on top of each other during training.
>
> * The relative speed-up of WarpNet compared to ResNet needs to be better explained — the authors break the computation of the WarpNet onto two GPUs, but it’s not clear if they do this for the (vanilla) ResNet as well. In batch mode, the easiest way to parallelize is to have each GPU evaluate half the batch. Even in a streaming mode where images need to be evaluated one by one, there are ways to pipeline execution of the residual blocks, and I do not see any discussion of these alternatives in the paper.
>
> We have performed experiments on ResNet with data parallelization using mini-batches. The results and discussion that compare WarpNet to ResNet with data parallelism are shown in Section 4.3.
>
> * In the experimental results, K is set to be 2, and the authors only mention in passing that they have tried larger K in the conclusion. It would be good to have a more thorough experimental evaluation of the trade-offs of setting K to be higher values.
>
> We have performed experiments with K=3. It is a much more interesting case than K=2, as there are 8 terms in the K=3 case and we only have 4 GPUs available. We made further approximations to the Taylor series by simply omitting terms in the Warp operator. Although in this case, WarpNet is not an exact first-order approximation of the ResNet. Results for K=3 are added in Section 4.1 (Table 3).
>
> * A major part of the paper tries to make the case that there is a symmetry breaking property of the proposed model, which I am afraid I simply was not able to follow. Some of the notation is confusing here — for example, presumably the rotations refer to image level rotations rather than literally multiplying the inputs by an orthogonal matrix, which the notation suggests to be the case.
>
> Sorry about the confusion. We decided to remove the discussion on symmetry breaking and use the space to show more extensive experimental results to demonstrate the effectiveness of WarpNet.

---

> > ### Author Response · Authors · 2018-01-03
> > **Response to AnonReviewer2 (cont'd)**
> >
> > *It is also never precisely spelled out what the final theoretical guarantee is (preferably the authors would do this in the form of a proposition or theorem).
> >
> > We have made our statements much clearer with Theorem 1 in Section 2, which shows the almost exact formula (without the epsilon^2 terms using ReLU). We provide two proofs of this theorem in appendix A.
> >
> > The first is a brute force approach, where we show that the number of terms with the same powers in F and F' after consecutive Taylor expansions satisfies the same recursion relation as the binomial coefficients. The also show explicitly that the number of terms with the same power in F and F' for x3 and x4 are binomial coefficients. Then, by induction, it follows that the number of terms of power k across K residual units is the binomial coefficient (K, k).
> >
> > The second proof starts by noting that each iteration of the Taylor series expansion can be described as a Bernoulli process with parameters K and p=0.5. It follows that any term in the Taylor series expansion is a realization of the Bernoulli process. The underlying Bernoulli variables correspond to whether a term gets an addition power of F' in the Taylor series expansion. Then the total number of terms with power k is the binomial coefficient (K,k).
> >
> > * There is a parallel submission (presumably by different authors called “Residual Connections Encourage Iterative Inference”) which contains some related insights. I wonder what are the differences between the two Taylor expansions, and whether the insights of this paper could be used to help the other paper and vice versa?
> >
> > Thanks for bringing this up. There are indeed differences in the two Taylor series expansions. We expand the outputs of residual units across layers where they expand the loss function. The two Taylor series behaves completely differently. Our Taylor expansion has almost vanishing second and higher order terms with ReLU non-linearity. However this does not appear to be guaranteed in their expansion.
> >
> > * On implementation - the authors mention using Tensorflow’s auto-differentiation. My question here is — are gradients being re-used intelligently as suggested in Section 3.1?
> >
> > Our experiments did not intelligently re-use the gradients as mentioned. It is a theoretical possibility that deserves to be looked at for possibly further speeding up WarpNet in a future investigation. We added the mentioning of this in the second paragraph of Section 4.
> >
> > * I notice that the analysis about the vanishing Hessian could be applied to most of the popular neural network architectures available now. How much of the ideas offered in this paper would then generalize to non-resnet settings?
> >
> > At least analytically, if ReLU is used, we can Taylor expand and just keep the first order terms in the series expansion. For ResNets, the Taylor expansion parameter is F. In general, however, the Taylor expansion parameter may be something else.

---

### Public Comment · (anonymous) · 2017-12-06
**Fatal mathematical flaw and weak experiments (part 2/2)**

"Therefore, a the first order term in the perturbation series is sufficient to give a good approximation across layers in ResNet."

No! In typical architectures, the first-order approximation of a deep ResNet is not close at all to the actual ResNet. This can be verified easily by simply computing that approximation directly. You will find that outputs are very different and the error of the approximation, while probably better than chance, will be nowhere near the original trained network.

"This is a good approximation, as the expected gradient norm across one residual unit was shown to be of order 10^-6"

No! The norm of the Jacobian of the residual path is ~ 1 / sqrt(i) in the typical architecture defined above if we ignore terms pertaining to network width. It is certainly nowhere near 10^-6. The Veit paper uses the value 10^-6 only once, but in a completely different context that this author claims. This shows that the authors of this paper have not understood the Veit paper.

"Therefore all second order perturbations vanish with probability 1. The same argument applies to higher orders. The first order perturbation theory is almost surely an exact theory for ResNets with ReLU non-linearity."

The fact that ReLU has no second derivative almost surely does not mean that the first order theory is exact. It is only exact in a region around each point that is enclosed by a non-differentiable boundary. But that region is much, much smaller than the size of the residual function and therefore does not apply to the author's analysis.

"This is an important result, which suggests that to the first non-vanishing order in gradient norm, the local minima is independent of parameters in subsequent residual units."

This is not true, because the O(\epsilon) assumption is not true.

"In principle, all layers in Resnet can be skipped and trained in parallel with the warp operator in WarpNet. As a proof of concept, we implemented a WarpNet that skips every other layer and we were able to obtain speedup."

No, WarpNet only works because it "parallelizes" only pairs of layers. If all layers were parallelized in this way, the network would perform very badly. Parallelizing pairs of layers is a relatively mild approximation and so it does not lead to catastrophic results because of the inherent robustness of neural networks.

Summary

This paper makes a number of incorrect statements. If it were accepted, it would greatly confuse readers that do not have a deep understanding of ResNet and would cause net damage to the community. This paper provides few experiments that show a mild improvement that is nowhere near sufficient to carry the paper by itself.

Beyond the issue discussed, this paper has several other significant weaknesses which I won't go into because I'm not an official reviewer and my time is limited. For questions / criticisms of this comment, please respond below. If the area chairs want to discuss directly, my identity is equal to that of AnonReviewer2 of the paper "Tandem Blocks in Deep Convolutional Neural Networks".

Confidence: 5 : This reviewer is absolutely certain.

---

> ### Author Response · Authors · 2017-12-08
> **No fatal mathematical flaws in our analysis (part 2)**
>
> "*No, WarpNet only works because it "parallelizes" only pairs of layers. If all layers were parallelized in this way, the network would perform very badly. Parallelizing pairs of layers is a relatively mild approximation and so it does not lead to catastrophic results because of the inherent robustness of neural networks."
>
> The approximation is almost exact with ReLU as both Veit's and our work together have shown, up to a path length of 25. We have performed "parallelization" over 3 layers in various settings in the past few weeks after obtaining access to servers with 4GPUs which allowed us to scale up our experiments. We found that the predictive accuracy remains similar to that of a ResNet with the same number of parameters, and the speedup of WarpNet over ResNet is increased when K=3 compared to K=2. ReLUs are used in our experiments.
>
> "*No! In typical architectures, the first-order approximation of a deep ResNet is not close at all to the actual ResNet. This can be verified easily by simply computing that approximation directly. You will find that outputs are very different and the error of the approximation, while probably better than chance, will be nowhere near the original trained network."
>
> The first order approximation is actually almost exact for the actual ResNet if ReLUs are used. We have analytically shown that the first order Taylor series expansion agrees with Veit's Figure 6(a) for the path length distribution, and 6(b) for the gradient scaling up to path lengths of about 25. Both Veit and we employed ReLUs. The validity of the linear approximation extends beyond the effective path length (Fig 6(c) in Veit) which is at most 20. This confirms that the first order series is almost exact.
>
> We guess that the confusion might be caused by our infrequent mentioning that ReLU is used in our analysis, and we will state that we have used ReLU non-linearities more often in our paper.
>
> "*No! The norm of the Jacobian of the residual path is ~ 1 / sqrt(i) in the typical architecture defined above if we ignore terms pertaining to network width. It is certainly nowhere near 10^-6. The Veit paper uses the value 10^-6 only once, but in a completely different context that this author claims. This shows that the authors of this paper have not understood the Veit paper."
>
> There are two (not only once) appearances of 10^-6 in Veit's paper. We took the 10^-6 number from Figure 6(b). There it shows that the gradient norm descents from about 10^-6 down to 10^-25 at path length 25. Another appearance of 10^-6 just before section 4.3 in Veit et al, we did not use that 10^-6 there.
>
> "*The fact that ReLU has no second derivative almost surely does not mean that the first order theory is exact. It is only exact in a region around each point that is enclosed by a non-differentiable boundary. But that region is much, much smaller than the size of the residual function and therefore does not apply to the author's analysis."
>
> Only a single point (the origin) in ReLU is non-differentiable. In any other region the first order term approximation is exact. As discussed before our work shows the first order approximation is almost exact with experimental validation from Veit et al.
>
> "*'This is an important result, which suggests that to the first non-vanishing order in gradient norm, the local minima is independent of parameters in subsequent residual units.'This is not true, because the O(\epsilon) assumption is not true."
>
> As we discussed before, the second order terms vanish because of the ReLU non-linearity we used. The O(\epsilon) number taken by the researcher is used for another purpose in another section of the paper

---

> > ### Public Comment · (anonymous) · 2017-12-14
> > **discussion continued above**
> >
> > I am continuing the discussion under the "Fatal mathematical flaw and weak experiments (1/2)" thread.

---

### Public Comment · (anonymous) · 2017-12-06
**Fatal mathematical flaw and weak experiments (1/2)**

Rating: 2/10

While the architecture presented (WarpNet) has slight performance improvements over comparable non-WarpNets in a few experiments, the paper is filled with incorrect statements which make it a clear reject.

First, let's look at a fairly typical residual network. Let x_{i+1} = h_i(x_i)  + F_i(x_i) where h_i is the identity and F_i(x_i) = Conv(ReLU(BN(x_i))). Assume that BN / Conv do not have trainable bias and variance parameter or, equivalently, the bias parameters are equal to 0 and the variance parameters are equal to 1, which they usually are in their initialized state. If the convolution is He-initialized and the input is normalized, then it is easy to check that ||h_i(x_i) || / ||F_i(x_i)|| ~ sqrt(i). This relationship does not change greatly throughout training. If we assume that the network has at most 100 residual blocks, which is often true in practice, the value of this ratio does not exceed 10. The same holds if F_i(x_i) = Conv(ReLU(BN(Conv(ReLU(BN(x_i)))))), another popular choice. Therefore, one of the central claims of the paper, which is that F_i and F_i' are very small, is false and this undermines the analysis presented.

Examples:

The paper states: "We now show that h must be close to the identity, up to O(\epsilon) << 1, when the output of a residual unit is similar to the input"

However we saw above that the output of a residual unit is usually at least 10% different from the input, so the assumption that the output of a residual unit is similar to its input up to O(\epsilon) << 1, is almost always false.

"F_i(x_i,W*_i) ~ O(e) from empirical observations such as those by Greff et al. (2016)"

We usually have ||F_i(x_i,W*_i)|| / ||x_i|| ~ sqrt(i), therefore F_i(x_i,W*_i) ~ O(e) usually does not hold. Also Greff et al did not observe F_i(x_i,W*_i) ~ O(e).

(3) is not meaningful. In practical ResNets, the O(e^2) term would dominate. While it is true that individual higher-order terms are smaller, this is more than made up by their greater frequency. In the Veit et al paper that this paper cites repeatedly, it is clearly demonstrated that in a typical ResNet, terms of a certain order dominate the gradient expansion (and therefore the Taylor expansion) and that this order is significantly greater than 1.


(comment continued below due to character limit.)

---

> ### Author Response · Authors · 2017-12-08
> **No fatal mathematical flaws in our analysis**
>
> Dear Anonymous Researcher,
>
> Thank you for your attention to our paper. We take all your points seriously, and do not find that our paper has fatal mathematical flaws. Rather we maintain that WarpNet is a sound and effective approximation of ResNet when ReLU is used. In addition, we have performed experiments for skipping over two layers in the past few weeks after we submitted the paper, and we found further speed up than just skipping one layer while maintaining similar predictive accuracy. We will add the new results into the paper before the revision deadline. Below we address your points one by one.
>
> "*Therefore, one of the central claims of the paper, which is that F_i and F_i' are very small, is false and this undermines the analysis presented."
>
> While it is true that ||F|/||h|| ~ 1/sqrt(i), the gradient norms ||F'|| are typically of order 10^-6 and smaller, see Figure 6(b) in Veit. Actually, the value of ||F||/||h|| has nothing to do with the validity of using the first order Taylor series to expand F (and thus the validity of the WarpNet). The purpose of the paragraph in question is an attempt to explain some observations about ResNet found in the literature. That paragraph in question only pertains to ResNet and we do not need small a ||F||/||h|| to perform the Taylor expansion on F. Therefore the subsequent sections and the validity of WarpNet are unaffected.
>
> That being said, we thank the researcher for pointing out ||F||/||h|| ~ 1/sqrt(i). We agree with you that we should not say F is very small. We will remove this confusing and irrelevant paragraph and it will certainly make our paper easier to read. It actually better motivates the expansion of F if F is not very small.
>
> "*(3) is not meaningful. In practical ResNets, the O(e^2) term would dominate. While it is true that individual higher-order terms are smaller, this is more than made up by their greater frequency. In the Veit et al paper that this paper cites repeatedly, it is clearly demonstrated that in a typical ResNet, terms of a certain order dominate the gradient expansion (and therefore the Taylor expansion) and that this order is significantly greater than 1."
>
> Regrettably, we are unable to find any form of series expansion in Veit's paper. We would appreciate it if the researcher would kindly point us to the location in Veit et al that they refer to as the "gradient expansion". The O(e^2) term does not dominate with ReLU non-linearities. Veit's paper confirms our analytical results.

---

> > ### Public Comment · (anonymous) · 2017-12-14
> > **Clarification questions**
> >
> > Dear authors,
> >
> > Thank you for your response to my comment. I apologize for the delay in getting back. I was revising my own paper and integrating reviewer requests. My response time should be much lower going forward.
> >
> > Before I respond to your latest comments, please answer the following three questions below. I want to exclude the possibility of a misunderstanding.
> >
> > - What is the exact form of equation (3) when the subscripts of F and W are added back in? Please provide a formula I can copy-paste into latex
> > - What is the exact formula of WarpNet if we didn't approximate groups of 2 layers, but arbitrarily large groups of layers? latex-formula and / or Pseudo-code would be great.
> > - What is your argument / evidence for the first-order approximation being exact for a ReLU network?
> >
> > Thanks,

---

> > > ### Author Response · Authors · 2018-01-03
> > > **Response to Anonymous Researcher**
> > >
> > > Dear Researcher,
> > >
> > > Thank you for your response. In the revised version of the paper we have addressed your new questions as follows
> > >
> > > - What is the exact form of equation (3) when the subscripts of F and W are added back in? Please provide a formula I can copy-paste into latex
> > >
> > > The equation (3) in page 3 in the revised paper now sums over all relevant subscripts of F and W.
> > >
> > > - What is the exact formula of WarpNet if we didn't approximate groups of 2 layers, but arbitrarily large groups of layers? latex-formula and / or Pseudo-code would be great.
> > >
> > > Equation (3) now is also valid for arbitrarily large group of layers. Also in Appendix A we spell out the formula for K=3. See Equation (9).
> > >
> > > - What is your argument / evidence for the first-order approximation being exact for a ReLU network?
> > >
> > > The argument is that when ReLU is used, the Hessian F''(x) vanishes almost exactly as ReLU is a piecewise linear function. This is given in the first paragraph of page 3. We also added experimental evidence in Figure 2, where it shows the validation curves of the wide residual network (blue solid) and its WarpNet (green dashed) approximation lie almost exactly on top of each other.

---

> > > > ### Public Comment · (anonymous) · 2018-01-12
> > > > **There is indeed a mathematical flaw and the experiments are still insufficient (part 5/5)**
> > > >
> > > > %##################
> > > > ++++++ Part (B): validity and merit of WarpNet
> > > > %##################
> > > >
> > > > As the authors pointed out, the statements made in sections 1 and 2 do not necessarily affect the merit of WarpNet. This is because the questions ``Does WarpNet approximate ResNet?'' and ``Can WarpNet learn successfully?'' are two different questions. We can view WarpNet simply as a model without considering its relation to ResNet and then consider its merit as a model.
> > > >
> > > > Unfortunately, the paper is also lacking in this area. The only postulated advantage of WarpNet over ResNet is the speedup obtained when parallelized. Firstly, not everyone has the resources to parallelize training. Secondly, in many cases those other GPUs are needed to evaluate other hyperparameter configurations. So the benefits of WarpNet are limited.
> > > >
> > > > Furthermore, WarpNet is likely over-complicated. Consider the WarpNet $x + F_1 + F_2 + F_2'F_1$. Why not use $x + F_1 + F_2 + F_2F_1$ instead? Or $x + F_1 + F_2$? I see no reason for the use of derivatives in the forward pass. Simpler models not using derivatives in the forward pass need to be shown to be inferior to WarpNet in order for WarpNet to have merit.

---

> > > > ### Public Comment · (anonymous) · 2018-01-12
> > > > **There is indeed a mathematical flaw and the experiments are still insufficient (part 4/5)**
> > > >
> > > > +++ Rebutting the new version of the paper
> > > >
> > > > Paper: The results imply that the output of a residual unit is just a small perturbation of the input.
> > > >
> > > > Response: The authors have themselves acknowledged that $\frac{||F_i||}{||h_i||} \sim \frac{1}{\sqrt{i}}$, so this statement is false.
> > > >
> > > > Paper: We find that merely the first term in the series expansion is sufficient to explain the binomial distribution of path lengths and exponential gradient scaling experimentally observed by Veit et al. (2016)
> > > >
> > > > Response: This statement makes no sense. The authors study the Taylor expansion, Veit studies the multiplied-out gradient. Those are different things.
> > > >
> > > > Paper: The approximation allows us to effectively estimate the output of subsequent layers using just the input of the first layer
> > > >
> > > > Response: The estimate is inaccurate to the point of being meaningless.
> > > >
> > > > Paper:  Below we show that the second and higher order terms are negligible, that is, the first order Taylor series expansion is almost exact, when ReLU activations are used. The second order perturbation terms all contain the Hessian $F''(x)$.
> > > >
> > > > Response: The fact that $F''$ is zero does not help. No matter how far you expand the Taylor series, you will always have an $O(.)$ residue, and that residue, in the case of ReLU will not shrink no matter how far you expand, and will lead to fatal inaccuracy.
> > > >
> > > > (3) is false because higher-order terms are not negligible.
> > > >
> > > > Paper: ``If one takes F to have ReLU non-
> > > > linearity, then $F''(x, W) = 0$ except at the origin. The non-trivial gradient can be expressed almost
> > > > exactly as  $K(F'(x, W))k$. This validates the numerical results that the gradient norm decreases k exponentially with subnetwork depth as reported in (Veit et al., 2016).''
> > > >
> > > > Response: This statement makes no sense, because the multiplied-out gradient does not contain $F''$ to begin with, so it is irrelevant that $F''$ vanishes.
> > > >
> > > > Paper: $\partial_{W_1} F_1 \approx 0$.
> > > >
> > > > Response: This is wrong on two levels. First, the authors seem to pre-suppose that $\frac{\partial x_3}{\partial W_1}$ is zero. This is not true. Even at a local minimum, only the derivative of the error with respect to $W_1$ is zero, but the derivative of $x_3$ with respect to $W_1$ is not necessarily zero. Secondly, $||F'_2|| \approx 10^{-6}$ is false as explained above.
> > > >
> > > > Paper: the local minima is independent of parameters in subsequent residual units.
> > > >
> > > > Response: The basic nature of deep networks is that parameters in each layer co-adapt to what parameters in other layers have learnt. This statement, if it were true, would contradict this basic nature.
> > > >
> > > > The equation pertaining to the gradient of the Taylor expansion at the top of page 4 is inaccurate because the Taylor expansion is inaccurate.

---

> > > > ### Public Comment · (anonymous) · 2018-01-12
> > > > **There is indeed a mathematical flaw and the experiments are still insufficient (part 3/5)**
> > > >
> > > > +++ Rebutting the authors second rebuttal (Title ``No fatal mathematical flaws in our analysis (part 2'')
> > > >
> > > > Author: The approximation is almost exact with ReLU as both Veit's and our work together have shown, up to a path length of 25. We have performed "parallelization" over 3 layers in various settings in the past few weeks after obtaining access to servers with 4GPUs which allowed us to scale up our experiments. We found that the predictive accuracy remains similar to that of a ResNet with the same number of parameters, and the speedup of WarpNet over ResNet is increased when K=3 compared to K=2. ReLUs are used in our experiments.
> > > >
> > > > Response: The approximation is not exact, as explained above. Also, the predictive accuracy has nothing to do with whether the approximation is exact (as I explain in Part B below).
> > > >
> > > > Author: The first order approximation is actually almost exact for the actual ResNet if ReLUs are used. We have analytically shown that the first order Taylor series expansion agrees with Veit's Figure 6(a) for the path length distribution, and 6(b) for the gradient scaling up to path lengths of about 25. Both Veit and we employed ReLUs. The validity of the linear approximation extends beyond the effective path length (Fig 6(c) in Veit) which is at most 20. This confirms that the first order series is almost exact.
> > > >
> > > > Response: Figure 6 of Veit has absolutely nothing to do with the accuracy of the Taylor expansion, neither does the superficial similarity of the Taylor expansion and gradient ``expansion'' have anything to do with the accuracy.
> > > >
> > > > Author: We guess that the confusion might be caused by our infrequent mentioning that ReLU is used in our analysis, and we will state that we have used ReLU non-linearities more often in our paper.
> > > >
> > > > Response: My criticisms have nothing to do with the fact that ReLU's were used. My criticisms are valid for all popular ResNet architectures.
> > > >
> > > > Author: There are two (not only once) appearances of $10^{-6}$ in Veit's paper. We took the $10^{-6}$ number from Figure 6(b). There it shows that the gradient norm descents from about $10^{-6}$ down to $10^{-25}$ at path length 25. Another appearance of $10^{-6}$ just before section 4.3 in Veit et al, we did not use that $10^{-6}$ there.
> > > >
> > > > Response: See prelim 2.
> > > >
> > > > Author: Only a single point (the origin) in ReLU is non-differentiable. In any other region the first order term approximation is exact. As discussed before our work shows the first order approximation is almost exact with experimental validation from Veit et al.
> > > >
> > > > Response: As I have shown in Prelim 1, even if the function is linear almost everywhere, the Taylor expansion is not necessarily exact.

---

> > > > ### Public Comment · (anonymous) · 2018-01-12
> > > > **There is indeed a mathematical flaw and the experiments are still insufficient (part 2/5)**
> > > >
> > > > +++ Rebutting the authors first rebuttal (Title "No fatal mathematical flaws in our analysis")
> > > >
> > > > Author: While it is true that $||F||/||h|| \sim 1/\sqrt{i}$, the gradient norms $||F'||$ are typically of order $10^{-6}$ and smaller, see Figure 6(b) in Veit.
> > > >
> > > > Response: As I explained in Prelim 2 and 3, the gradient norms are of the same order as the forward norms, and figure 6(b) of Veit does not mean what you think it means.
> > > >
> > > > Author: Actually, the value of $||F||/||h||$ has nothing to do with the validity of using the first order Taylor series to expand F.
> > > >
> > > > Response: In prelim 1, I showed how the Taylor expansion can be highly inaccurate. In your paper, you expand functions around $h$ with perturbation $F$. So the larger the perturbation, the greater the inaccuracy of the Taylor expansion tends to be. Therefore the value of $||F||/||h||$ matters.
> > > >
> > > > Author: we do not need small a $||F||/||h||$ to perform the Taylor expansion on $F$
> > > >
> > > > Response: As mentioned above, if the expansion is to be accurate and thus meaningful, the perturbation must be small.
> > > >
> > > > Author: Regrettably, we are unable to find any form of series expansion in Veit's paper.
> > > >
> > > > Response: By ``expansion'' I simply mean multiplying out the gradient $(h'_1 + F'_1)(h'_2 + F'_2)..$ into its $2^L$ components.
> > > >
> > > > Author: The $O(e^2)$ term does not dominate with ReLU non-linearities. Veit's paper confirms our analytical results.
> > > >
> > > > Response: The $O(e^2)$ term does dominate, and this is the crucial point. In Prelim 1, I showed how the first-order Taylor expansion can be highly inaccurate even if the function is piecewise linear. I used a ReLU-like function $\max(100x,0)$ as an example. Now, of course the inaccuracy for a regular ReLU, i.e. $\max(x,0)$ is going to be less. However, if you expand the second layer with respect to the first, and then the third layer with respect to that, and so forth, eventually those inaccuracies will compound and eventually dominate the approximation.
> > > >
> > > > I have no idea why you think Veit confirms the result you claim. The Veit paper did not even consider the Taylor expansion. The only expansion it contains is the one I referred to above, the multiplying out of the gradient. However this ``expansion'' is exact, because multiplying out is exact, as opposed to Taylor.

---

> > > > ### Public Comment · (anonymous) · 2018-01-12
> > > > **There is indeed a mathematical flaw and the experiments are still insufficient (part 1/5)**
> > > >
> > > > %My rebuttal can be pasted an viewed in Latex
> > > >
> > > > %If you want to respond to this rebuttal, please respond with a single post or a single block of posts below the last part of this comment (part X).
> > > >
> > > > %
> > > > %
> > > > %
> > > >
> > > > The authors rebuttal and the updated version of the paper confirm that the authors have fundamental misconceptions about ResNet, Veit's work and the Taylor expansion which leads to many false, misleading and / or meaningless statements in the paper, as I show below in Part A.
> > > >
> > > > The authors are correct in saying that those misconceptions and false statements do not necessarily affect the validity / merit of WarpNet. Evaluating the merit of WarpNet on its own terms is a worthwhile exercise. I will do this below in Part B. However, it turns out that the WarpNet model itself also lacks merit. The arguments presented in parts A and B independently require the paper to be rejected.
> > > >
> > > > %##################
> > > > ++++++ Part (A): misconceptions and false statements
> > > > %##################
> > > >
> > > > +++ Prelim (1): The Taylor expansion.
> > > >
> > > > The basic form of the Taylor expansion is: $f(x + e) = f(x) + ef'(x) + O(e^2)$, where $f$ is differentiable at $x$. The definition of the $O(e^2)$ is `a quantity that when divided by $e^2$ is bounded as $e$ converges to zero'. So the only thing we know about $O(e^2)$ is its behavior as $e$ tends to zero, but we don't know anything about what its value is for a given value of $e$.
> > > >
> > > > Consider the function $f(x) = \max(0, 100x)$. Let $x = -0.1$ and $e=0.2$. Then we have $f(x + e) = 10$, but the Taylor expansion yields the value 0. So even though $e$ is quite small and $f$ is linear everywhere except at one point, the Taylor expansion is very inaccurate.
> > > >
> > > > +++ Prelim (2): Figure 6(b) of Veit et al.
> > > >
> > > > This figure depicts the magnitude of the gradient along certain paths as a function of the number of residual derivatives contained in that path (i.e. the number of $F'$ terms contained in it). In this figure, we find that the path containing no residual derivatives has a magnitude of $\approx 10^{-5}$ whereas the average path containing 30 residual derivatives has a magnitude of $\approx 10^{-23}$. This tells us that each individual residual derivative has a size of around $\big(\frac{10^{-23}}{10^{-5}}\big)^{-30} \approx 0.25$. (CORRECTION: The formula should be $\big(\frac{10^{-23}}{10^{-5}}\big)^{\frac{1}{30}} \approx 0.25$.) Assuming that no other scaling effects were involved in the creation of this graph, the fact that the path containing no residual derivatives has a magnitude of $\approx 10^{-5}$ also tells us that the derivative of the error function had size of $\approx 10^{-5}$, as it is the only derivative contained in that path.
> > > >
> > > > +++ Prelim (3): Explaining ResNet
> > > >
> > > > As I previously explained and as the authors acknowledged, in a batch-ReLU ResNet we have $\frac{||F_i||}{||h_i||} \sim \frac{1}{\sqrt{i}}$. But crucially we also have $\frac{||F_i||}{||h_i||} \approx \frac{||F'_i||}{||h'_i||}$. This is because all operations involved in $h$ and $F$ are very similar when evaluated in the forward and backward direction.
> > > >
> > > > \begin{itemize}
> > > > \item The identity function $h(x) = x$ is equivalent to multiplication with the identity matrix in both directions.
> > > > \item A linear transformation $Wx$ is equivalent to multiplication with $W$ in both directions.
> > > > \item ReLU, in both directions, is equivalent to multiplication with the same binary matrix of 0's and 1's depending on which neurons are activated.
> > > > \item While batchnorm only centers the mean in the forward directions, the multiplicative effect is the same in both directions.
> > > > \end{itemize}
> > > >
> > > > Hence, we also have $\frac{||F'_i||}{||h'_i||} \sim \frac{1}{\sqrt{i}}$. In a 56-block ResNet as used by Veit, the average of all $\frac{1}{\sqrt{i}}$ values is $\frac{1}{56}\sum_1^{56} \frac{1}{\sqrt{i}} \approx 0.24$. And now we come full circle by realizing that $0.24 \approx 0.25$ so this analysis of ResNet matches Veit's results.

---

> > > > > ### Author Response · Authors · 2018-01-13
> > > > > **Taylor expansion is accurate and WarpNet is sound, innovative and effective**
> > > > >
> > > > > Dear Researcher,
> > > > >
> > > > > We think that the debate between the Researcher and us rises partially from a misunderstanding from the Researcher's part on when one can apply the Taylor expansion.
> > > > >
> > > > > The Taylor expansion can only be applied to provide a LOCAL polynomial type of approximation of a function value at a relatively small neighborhood of a given point which does not lie on any boundary. Mathematically speaking, a Taylor expansion should only be done in a small neighborhood of a given x, say (x-e,  x+e), where e is a small numerical term.
> > > > >
> > > > > Many of the Researcher's arguments are based on a constructed example in Prelim 1) which tried to demonstrate the inaccuracy of the Taylor expansion. However, that example is irrelevant since a Taylor expansion should NOT be applied in the first place. In particular, e is much bigger than x so it violates the condition of using Taylor expansion. It is akin to forcing a short term weather forecast model to do a long term prediction.
> > > > >
> > > > > In our framework, all conditions of applying the Taylor expansion have been checked carefully. In our case of F2(x1+F1), x = x1 and e = F1. It is clear that F1 < x1.
> > > > >
> > > > > We find that the Taylor expansion is not only suitable but also brings very significant computational advantages as explained in the next paragraph.
> > > > >
> > > > > The WarpNet (from the Taylor expansion) provides a novel way of training ResNets in parallel: model parallelism in which different sets of weights are trained in parallel (which we believe is attractive to the industry). It can be trained faster compared to mini-batch parallelization of ResNet (Section 4.3), and allows a larger network to be trained (for which ResNet fails to train due to GPU memory limitation, see Section 4.2) as WarpNet splits the weight storage into different GPUs. This offers significant advantages since GPU memory is often quite limited. Our experimental results also show that the predictive performance of WarpNet and that of the corresponding ResNet are very similar owing to the good Taylor expansion. Further, we find that the validation error curve of WarpNet can lie on top of each other with that of ResNet throughout training (Figure 2).
> > > > >
> > > > > After carefully reading the Researcher's responses, it appears that some of the disputes arise from different definitions of "high order terms". For instance, in Part [2/5], the Researcher mentions that $O(e^2)$ terms dominate, and that "While it is true that individual higher-order terms are smaller, this is more than made up by their greater frequency." in an earlier response. We conclude that the "high order terms" the Researcher refer to is not the same as our "high order terms".
> > > > >
> > > > > To be more specific, the Researcher appears to refer to the binomial number of terms in powers of $F'$ by multiplying out the gradient $(h'_1 + F'_1)(h'_2 + F'_2)..$ in a ResNet. Following our convention in the paper, we denote these as binomial terms. However, our "high order terms" correspond to the order of derivative terms in the Taylor expansion. The binomial terms and the Taylor series terms are completely different. In our analytical analysis we included -all- binomial terms, which the Researcher refers to as "high order terms". The Taylor series is truncated by the ReLU to first order. However, in Appendix A we have shown that using multiple iterations of Taylor expansions layer by layer results in a binomial number of first order terms, where each binomial term consists of a product of $k-1$ $F'$. Nowhere in the analytical analysis we have neglected first order Taylor series terms of power larger than 1 in $F'$. We believe that this resolves the discrepancy between our results and those stated by the Researcher.

---

> > > > > > ### Author Response · Authors · 2018-01-14
> > > > > > **Response to the Researcher's other comments**
> > > > > >
> > > > > > We have studied the Researcher's Prelims (2) and (3). We do not think the formula in the middle of Prelim (2), that is, $\big(\frac{10^{-23}}{10^{-5}}\big)^{-30} \approx 0.25$, holds. We doubt about Prelim (3) as well.
> > > > > >
> > > > > > We find that the statement made by the Researcher in prelim 3 does not seem to be qualitatively correct. For instance, it is stated that "This is because all operations involved in $h$ and $F$ are very similar when evaluated in the forward and backward direction." However, it can be seen that ||h'|| is the norm of the kronecker delta, drastically different when compared to ||h||. Further, in F', the non-linearity layers have inherently different shapes - for sigmoid and tanh, the forward pass is only zero at the origin, whereas in the backward pass, the gradient of sigmoid and tanh are essentially zero in the flat region of sigmoid and tanh away from the origin. Similarly for ReLU, the gradient is the step function, and only has similar behaviors near values h = 1. F and F' behaves differently on a qualitative level.
> > > > > >
> > > > > > Therefore, it does not appear that this is the correct explanation ||F'||/||h'|| has a similar value to ||F||/||h||, as F and F' behave drastically differently in backward and forward passes. We think that this is overreaching on a scale larger than us ignoring the second term in the gradient decoupling section even if ||F'|| is of order O(0.1).
> > > > > >
> > > > > > In addition, our discussion on gradient decoupling in Section 2 is just to provide an analysis on ResNet. In WarpNet, we did not use gradient decoupling in the backward pass as shown in Section 3.2. To prevent the readers from thinking that we used gradient decoupling in the backward pass of WarpNet, we think we should drop off the paragraphs related to gradient decoupling in Section 2 (which only pertains to ResNet).
> > > > > >
> > > > > > Since the Researcher's Prelims (2) and (3) are only related to our discussions on gradient decoupling, this treatment resolves our dispute in this regard.
> > > > > >
> > > > > > Regarding the Researcher's comment “the experiments are still insufficient”, we would like to emphasize that in the revision we provided new experimental results from over 110 runs. The new experiments were done with different parameter settings on 3 data sets. We also compared with data parallelism. We are surprised to discover that the Researcher insists that the experiments are insufficient. We worked extremely diligently in the past two months to obtain these new results, and our results confirmed that WarpNet is an effective and much faster alternative to ResNet. We hope our innovative contribution can be properly recognized!
> > > > > >
> > > > > > Regarding the Researcher's comment "not everyone has the resources to parallelize training", now it is quite common for deep learning researchers and AI companies to have or have access to GPU servers with multiple GPUs. Companies that provide GPU cloud services (such as Google, IBM, Amazon, Oracle and Nvidia) have hundreds of GPUs in their cloud. We believe our proposed WarpNet will be very useful to researchers and practitioners who look for fast deep learning solutions.

---

> > > > > > > ### Public Comment · (anonymous) · 2018-01-17
> > > > > > > **Final rebuttal**
> > > > > > >
> > > > > > > First of all, apologies to the area chair(s) for the lengthy exchange. I'm not trying to cause unnecessary work or be confrontational. Since I decided to post an full review for the paper, I also decided to take on the same responsibility that I would take on as an official reviewer of the paper, which is to keep rebutting as long as new arguments are presented by the authors.
> > > > > > >
> > > > > > > On to the rebuttal.
> > > > > > >
> > > > > > > .
> > > > > > > .
> > > > > > > .
> > > > > > >
> > > > > > > Firstly, I made a typo in Prelim (2). I apologize. The correct formula is $\big(\frac{10^{-23}}{10^{-5}}\big)^{\frac{1}{30}} \approx 0.25$ instead of $\big(\frac{10^{-23}}{10^{-5}}\big)^{-30} \approx 0.25$. However, the point I was making very much stands. If a path containing 30 F' terms has size 10^-23 and a term containing one F' term has size 10^-5, then the typical F' term has size around 0.25, because 0.25^30 * 10^(-5) ~ 10^(-23)
> > > > > > >
> > > > > > > .
> > > > > > > .
> > > > > > >
> > > > > > > As the authors pointed out, Prelim (1) is an example of where Taylor should not be used. That is precisely why I gave the example, to show why Taylor should not be used for ResNet in the way the authors are using it. In the max(0, 100x) example I gave with x = -0.1 and e=0.2, x < 0 and so the max() selects the 0 term. But x + e > 0, so max() selects the 100x term, and so the Taylor expansion is inaccurate. The inaccuracy is caused by the Taylor expansion only being aware of max(0,100x) locally at the location x=-0.1, so it cannot distinguish max(0,100x) from the zero function. This shows that when using Taylor, any sudden changes to the function between x and x + e leads to inaccuracy.
> > > > > > >
> > > > > > > This is precisely what happens in ResNet! The authors expand, for example, F_2(x + F_1) around x. As we established long ago, ||F_i||/||h_i|| ~ 1/sqrt(i), so in this case ||F_1||/||x|| ~ 1/sqrt(1) = 1. Therefore, x and F_1 are of similar size. Let's assume we feed x into F_2. When we get to the ReLU layer, some ReLU's will not be activated (input < 0) and some ReLU's will be activated (input > 0). Now assume we feed x + F_1 into F_2. Since F_1 is of similar size to x, the values that go into the ReLU units will be substantially different. Therefore the effect I described in Prelim (1) will arise and a good number of ReLU inputs will switch from < 0 to > 0 and vice versa. Therefore, the Taylor approximation, which feeds x into F_2 instead of x + F_1 will incur an error. Now, that error will not be as big as in my Prelim (1) example where I used max(0,100x), but it will be significant.
> > > > > > >
> > > > > > > So, when we replace F_2(x + F_1) with F_2(x) + F_1 + F_2'F_1, there will be a small, but not insignificant error. Now consider F_3. In reality, F_3 is applied to x + F_1 + F_2. In the Taylor expansion, F_3 is applied to x. Now the difference between x and x + F_1 + F_2 is even greater than the difference between x and x + F_1, so values fed into ReLU are more likely to flip from > 0 to < 0 and vice versa, so the inaccuracy of Taylor is larger. Now consider F_n. In reality, it is applied to x + F_1 + F_2 + .. + F_n-1. In the Taylor setting, it is applied to x. But F_1 + F_2 + .. + F_n-1 dominate x in magnitude, so the ReLU activation pattern will be completely different, and the value of the Taylor approximation will be completely different from the true value. Therefore, the Taylor errors grow from layer to layer and also compound in the sense that they are added together. When we get to the networks output layer, those errors dominate. Therefore the theory presented by the authors in section 2, and formular (3) in particular, is incorrect.
> > > > > > >
> > > > > > > .
> > > > > > > .
> > > > > > >
> > > > > > > In the authors latest rebuttal, they discuss my use of the phrase "higher order terms". What I meant by that term is not what they claim I meant, but in the interest of brevity. In that rebuttal, they also question my Prelim (3) from my previous rebuttal. I maintain that this analysis is correct. In the interest of brevity, I will not go into those points here. I think my earlier posts speak for themselves. If the area chairs want me to explain further, they can contact me via email. My identity is equal to that of AnonReviewer2 of the paper "Tandem Blocks in Deep Convolutional Neural Networks".
> > > > > > >
> > > > > > > .
> > > > > > > .
> > > > > > >
> > > > > > > Finally, when I say the ``experiments are insufficient'', I don't mean in terms of the number of experiments. You could run experiments on a hundred different datasets, but one question remains: Why should I prefer, say, x + F_1(x) + F_2(x) + F'_2(x)F_1(x) to x + F_1(x) + F_2(x) or to x + F_1(x) + F_2(x) + F_1(x)F_2(x). Those are much simpler models that achieve the exact same objective of decoupling. The complexity of using derivatives in the forward pass is unjustified. Again, I'm not saying that WarpNet is unsound in the sense that it is an ineffective model. I am simply saying that the same benefits of WarpNet can likely be achieved without derivatives in the forward pass. If you want to advocate for derivatives in the forward pass, you need to show that they are necessary. Since you did not do that, the experiments are insufficient in that sense.

---

> > > > > > > > ### Author Response · Authors · 2018-01-17
> > > > > > > > **Response to Final Rebuttal**
> > > > > > > >
> > > > > > > > As we mentioned before, your statements in Prelims (2) and (3) are not relevant to WarpNet. Both our forward and backward passes are based on the result of the Taylor expansion. Your argument in Prelim (1) is also not valid as we pointed out in our last reply and below. We would like to reiterate that our experiments have confirmed that our framework indeed works.
> > > > > > > >
> > > > > > > > The general expression of the function is F_(i+1)(x_i+F_i). Suppose ||Fi||/||x_i|| ~ 1/sqrt(i). ||Fi||< ||xi|| for layers beyond the first one. Your example in Prelim (1) is far off for any of the layers (including the first one), and cannot be used to illustrate anything relevant to our approximation with the Taylor expansion.
> > > > > > > >
> > > > > > > > The whole point of doing the experiments is to see whether the approximation is good, and we demonstrated that the Taylor expansion leads to a good approximation to ResNet with much shorter training time.
> > > > > > > >
> > > > > > > > There are certainly ways to further approximate WarpNet or ResNet. The main purpose of our experiments is to illustrate the effectiveness of WarpNet compared to ResNet and data-parallelized ResNet.  We also provided the results of one approximation to WarpNet. There are certainly other ways to further approximate WarpNet or ResNet. But it is beyond the topic of this paper. The reason we have F’ in the forward pass is due to the Taylor expansion. The two approximations you mentioned do not have a theoretical basis.

---

> > > > > > > > > ### Public Comment · (anonymous) · 2018-01-18
> > > > > > > > > **Some Title**
> > > > > > > > >
> > > > > > > > > "Your argument in Prelim (1) is also not valid as we pointed out in our last reply and below. We would like to reiterate that our experiments have confirmed that our framework indeed works."
> > > > > > > > >
> > > > > > > > > Your framework "working" has absolutely nothing to do with whether the Taylor expansion is accurate or not. My main problem is that equation (3) in the paper is wrong and therefore may fundamentally confuse readers about the nature of ResNets. However, even if (3) is false and WarpNet does not approximate ResNet, that doesn't mean WarpNet isn't trainable. Here are some things that are not an approximation of a given ResNet: vanilla nets, decision trees, kernel machines, ResNet's with different initial weights, ResNet's with different nonlinearities. Yet these models can all be trained. So the argument "WarpNet can be trained, therefore it produces the same outputs as ResNets" is nonsensical.
> > > > > > > > >
> > > > > > > > > Prelims (1)-(3) are all true and explain in detail the Taylor approximation is inaccurate.
> > > > > > > > >
> > > > > > > > > The two alternative models I mentioned don't have have a theoretical basis, yes, but neither has WarpNet, because the Taylor expansion doesn't work.

---

> > > > > > > > > > ### Author Response · Authors · 2018-01-18
> > > > > > > > > > **Critical misunderstanding of our approach**
> > > > > > > > > >
> > > > > > > > > > Our formula (3) and its proof are correct and sound. We believe that your conclusion is based on a critical misunderstanding of our approach. As evident from the derivation of Equation 9 from Equation 8, our approximation is built upon MANY local Taylor expansions to build an approximation for each layer.
> > > > > > > > > >
> > > > > > > > > > Your argument is based on your misunderstanding that we tried to expand F3(x+F1+F2) or Fn(x+F1+F2+…+F_n-1) using only ONE Taylor expansion as indicated in one of your responses. This is not what we did at all. Thus, we would like to say again that your example and statements in Prelim (1) are not valid, and misleading. WarpNet has a solid theoretical basis, and is demonstrated in our experiments to be a good approximation to ResNet.

---

### Decision · Program_Chairs · 2018-01-29
**ICLR 2018 Conference Acceptance Decision**

**Decision:**

Accept (Poster)

**Comment:**

This paper proposes a “warp operator” based on Taylor expansion that can replace a block of layers in a residual network, allowing for parallelization. Taking advantage of multi-GPU parallelization the paper shows increased speedup with similar performance on CIFAR-10 and CIFAR-100. R1 asked for clarification on rotational symmetry. The authors instead removed the discussion that was causing confusion (replacing with additional experimental results that had been requested). R2 had the most detailed review and thought that the idea and analysis were interesting. They also had difficulty following the discussion of symmetry (noted above). They also pointed out several other issues around clarity and had several suggestions for improving the experiments which seem to have been taken to heart by the authors, who detailed their changes in response to this review. There was also an anonymous public comment that pointed out a “fatal mathematical flaw and weak experiments”. There was a lengthy exchange between this reviewer and the authors, and the paper was actually corrected and clarified in the process. This anonymous poster was rather demanding of the authors, asking for latex-formatted equations, pseudo-code, and giving direction on how to respond to his/her rebuttal. I don't agree with the point that the paper is flawed by "only" presenting a speed-up over ResNet, and furthermore the comment of "not everyone has access to parallelization" isn’t a fair criticism of the paper.